# Liquid metal-embraced photoactive films for artificial photosynthesis

Chao Zhen [1,11], Xiangtao Chen[2,11], Ruotian Chen[3], Fengtao Fan [3], Xiaoxiang Xu [4], Yuyang Kang[1], Jingdong Guo[1], Lianzhou Wang [5], Gao Qing (Max) Lu[6], Kazunari Domen [7,8], Hui-Ming Cheng[1,9] & Gang Liu [1,10] ✉

The practical applications of solar-driven water splitting pivot on significant advances that enable scalable production of robust photoactive films. Here, we propose a proof-of-concept for fabricating robust photoactive films by a particle-implanting technique (PiP) which embeds semiconductor photo-absorbers in the liquid metal. The strong semiconductor/metal interaction enables resulting films efficient collection of photogenerated charges and superior photoactivity. A photoanode of liquid-metal embraced $BiVO_4$ can stably operate over 120 h and retain ∼70% of activity when scaled from 1 to 64 $cm^2$. Furthermore, a Z-scheme photocatalyst film of liquid-metal embraced $BiVO_4$ and Rh-doped $SrTiO_3$ particles can drive overall water splitting under visible light, delivering an activity 2.9 times higher than that of the control film with gold support and a 110 h stability. These results demonstrate the advantages of the PiP technique in constructing robust and efficient photoactive films for artificial photosynthesis.

Direct solar-driven water splitting by photocatalysis or photoelectrolysis offers a very promising route to produce green hydrogen[1–6]. Among different configurations of photoactive materials, the bioinspired concept of "Z-scheme photoactive film" enables visible-light-driven water splitting by combining two different photoactive materials in series to separately produce $H_2$ and $O_2$ in tandem photoelectrochemical (PEC) cells[7–10] or photocatalytic panels has demonstrated its great effectivenss[11–13]. Their scale-up and commercial applications though require low-cost, stable, efficient, and scalable photoactive films, which are the essential components of these photons-driven water splitting systems[4,5]. Therefore, the desirable photoactive films shall be fabricable from scalable processing techniques, avoiding the use of scarce and expensive elements and vacuum fabrication[14–18] or

hydrothermal processes[19,20]. Current techniques either suffer low cost-effectiveness or have scale-up issues. Therefore it is of great need and interest to develop alternative techniques for photoactive film processing.

Although some chemical methods have been developed to produce various visible light responsive semiconductor photoactive films, including n-type $BiVO_4$[7,21,22], $Fe_2O_3$[8,9,23], $Ta_3N_5$[20], and p-type $Cu_2O$[10,24], $Sb_2Se_3$[25], $Cu_2ZnSnS_4$[26] semiconductors, post-treatments are generally indispensable to obtain the desired PEC performance, thus adding cost and complexity to these processes. Moreover, these post-treatments apparently increase the risk of photoelectrode heterogeneity, contributing to a marked decay of PEC activity when scaling up the working area, which has been frequently encountered in promising

[1]Shenyang National Laboratory for Materials Science, Institute of Metal Research, Chinese Academy of Sciences, 72 Wenhua Road, Shenyang 110016, China. [2]Key Laboratory for Anisotropy and Texture of Materials (Ministry of Education), Northeastern University, Shenyang, Liaoning 110819, China. [3]State Key Laboratory of Catalysis, Dalian National Laboratory for Clean Energy, iChEM, Dalian Institute of Chemical Physics, Chinese Academy of Sciences, Dalian, China. [4]School of Chemical Science and Engineering, Tongji University, Shanghai 200092, China. [5]Nanomaterials Centre, School of Chemical Engineering and AIBN, The University of Queensland, St Lucia, Brisbane, QLD 4072, Australia. [6]University of Surrey, Guilford GU2 7XH, UK. [7]Research Initiative for Supra-Materials, Shinshu University, Nagano, Japan. [8]Office of University Professors, The University of Tokyo, Tokyo, Japan. [9]Shenzhen Institute of Advanced Technology, Chinese Academy of Sciences, 1068 Xueyuan Blvd, Shenzhen 518055, China. [10]School of Materials Science and Engineering, University of Science and Technology of China, 72 Wenhua Road, Shenyang 110016, China. [11]These authors contributed equally: Chao Zhen, Xiangtao Chen. ✉e-mail: gangliu@imr.ac.cn

BiVO$_4$-based photoanodes[27–29]. In addition, it lacks a feasible strategy to fabricate multi-element semiconductor photoactive films (e.g., BaTaO$_2$N[30] and Y$_2$Ti$_2$O$_5$S$_2$[31]) owing to their harsh synthesis conditions.

Alternatively, a low-cost and facile processing route is to deposit semiconductor particles onto conductive substrates[32–35]. However, the resultant semiconductor films are generally fragile and of low activity in performance due to the weak physical adhesion between the deposited particles and the substrate, particularly for the large-sized particles. Although the attractive particle transfer method[36–39] has been developed to overcome these issues and further extended to the construction of photocatalytic panels for overall water splitting, in which a metal film is deposited on a film of particles under vacuum and then peeled off together with a layer of particles, the involvement high vacuum and gold supporter greatly increase the cost for scale-up manufacturing.

Inspired by the cellular structure of the thylakoid membrane of chloroplasts in plant cells[40,41], embedding pigment-protein complexes (Photosystem I/II) in lipid matrix (Supplementary Fig. S1) for efficient photosynthesis, in this study, a facile and scalable particle-implanting (PiP) fabrication technique of embedding semiconductor particles in low-melting-point (LMP) liquid metal matrixes was developed to produce bioinspired photoactive films. Compared with the photoactive films assembled on the surface of conductive substrates, the resulting unique film delivers drastically improved PEC water-splitting performances because the strong embedding-type metal/semiconductor interface enables high adhesion strength, strong photocarrier collection, and excellent activity-uniformity. Furthermore, a bioinspired

photoactive film was produced by using the liquid metal-based PiP technique with semiconductor particles of Rh-doped SrTiO$_3$ and BiVO$_4$ simultaneously embedded, which perform as photocatalysts for hydrogen evolution reaction (HER) and oxygen evolution reaction (OER), respectively. Upon visible light (≥420 nm) irradiation, the integrated panel system can stably produce stoichiometric H$_2$ and O$_2$ during a 110 h long operation. This work demonstrates the universality, scalability, and superiority of the liquid-metal-based PiP technique in constructing efficient, robust photoactive films for artificial photosynthesis.

## Results and discussion

### Proposed proof-of-concept for constructing bioinspired photoactive films

We developed a brand-new PiP technique of embedding various semiconductor particles in LMP liquid metals for the facile fabrication of bioinspired photoactive films (Fig. 1a). At a low temperature (usually above 50 °C), the LMP metal melts into a liquid that is of high flowability and conductivity, serving as an ideal substrate to anchor semiconductor particles for the formation of films. In this design, an LMP metal (e.g., the Field's metal) is melted above its melting point (62 °C) and coated on a substrate to form a liquid metal film. In response to the fact that the homogeneous coating of melted Field's metal on substrates is relatively difficult due to its high surface tension, we adopted a blade to rub the liquid metal on the support substrates for homogeneous surface wetting without any surface treatment (Supplementary Fig. S2). Subsequently, semiconductor particles are introduced

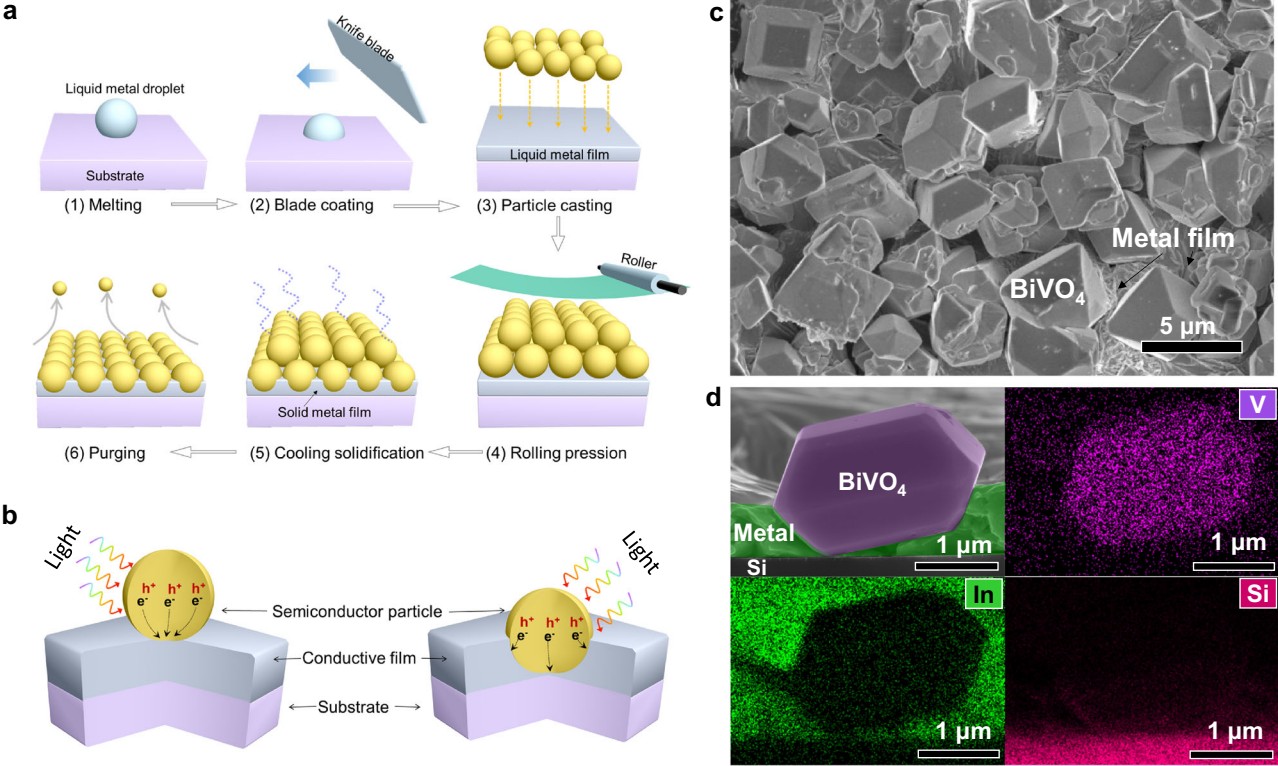

**Fig. 1 | Particle implanting process for the fabrication of bioinspired photoelectrode film. a** The fabrication process of bioinspired semiconductor photoelectrode using an LMP metal, including (1) metal melting and (2) blade coating on a substrate to form a liquid metal film, (3) semiconductor particles (indicated by yellow spheres) casting on its surface, (4) roller-pressing the semiconductor particles into the liquid metal film, (5) liquid metal film solidification with cooling, and (6) achieving a monolayer-particle-embedded photoelectrode film by blowing off the loose particles on film surface. **b** Typical structure of a particle-post-assembly photoelectrode obtained from conventional methods

(semiconductor particles deposited the conductive substrate in point-to-face contact manner) and the new type of bioinspired photoelectrode proposed in this work (semiconductor particles are embedded in the conductive metal layer and form three-dimensional intimate interfacial contact with the conductive layer). **c** The top-viewed scanning electron microscopy (SEM) image of the fabricated BiVO$_4$ photoelectrode using LMP metal. **d** The cross-sectional SEM image of the resultant BiVO$_4$ photoelectrode and the EDS mappings of Si, In, and V recorded on the cross-sectional SEM image.

onto its surface and then pressed into the liquid metal film which is then cooled for solidification to form a pebble-textured monolithic film. The loosely attached semiconductor particles on the film surface can be readily removed from the composite film by gas blowing and collected for reuse. In contrast to conventional photoactive films where particles and substrate are contacted in a point-to-face manner (left panel of Fig. 1b), the resulting photoactive film has implanted (or semi-embedded) semiconductor particles and a unique three-dimensional (3D) semiconductor/metal interface (right panel of Fig. 1b). Consequently, the photocarrier collection efficiency is expected to be substantially improved due to the significantly shortened carrier transport distance and drastically increased carrier collection paths. One can envision that, with an LMP metal as the conductive layer to anchor particles, different semiconductor particles can be selectively assembled onto various substrates using this PiP technique, affording it a generic approach for the scale-up manufacturing of photoactive films.

## Fabrication and characterization of bioinspired BiVO₄ photoactive film

Semiconductor metal oxides are generally considered to be of high stability among various photoactive materials, and $BiVO_4$ has been regarded as one of the most promising photoactive materials for water oxidation due to its appropriate bandgap and relatively low photocurrent onset potential[42]. Accordingly, $BiVO_4$ prevails over other semiconductors in tandem PEC cells and photocatalytic panels[5]. Here, we used faceted $BiVO_4$ as a model photoabsorber to fabricate the photoactive film as a proof-of-concept. The $BiVO_4$ particles prepared from a hydrothermal method are confirmed to have the monoclinic phase according to its XRD patterns (Supplementary Fig. S3a). SEM observation shows that the $BiVO_4$ particles have a truncated tetragonal bipyramid shape with a particle size of a few micrometers (Supplementary Fig. S3b). The well-developed top and side surfaces of the truncated tetragonal bipyramids are geometrically defined as the {010} and {011} facets of monoclinic $BiVO_4$[43]. The as-synthesized yellowish $BiVO_4$ powders (inset image Supplementary Fig. S3b) were first well dispersed in isopropanol solvent with the assistance of ultrasonication for use in the subsequent photoelectrode film assembly. We first chose an ultra-smooth Si wafer as the substrate for the integration of $BiVO_4$ particles. The $BiVO_4$ particles within the resultant photoactive film are uniformly anchored on the surface of the solidified Field's metal film in a monolayer manner (Fig. 1c). The cross-section SEM images clearly show that micrometer-sized $BiVO_4$ particles were embedded in the solidified metal film (Fig. 1d and Supplementary Fig. S4), affirming the effectiveness of the PiP technique. Energy dispersive X-ray spectroscopy (EDS) mappings confirm the locations of $BiVO_4$ particles, Field's metal, and Si substrate, further verifying the effective embedment of $BiVO_4$ particles in the Field's metal film according to the spatially complementary distributions between V (or O) and In (or Sn) elements (Fig. 1d and Supplementary Fig. S5). To conclusively demonstrate the formation of the intimate contact across the semiconductor/metal interface, we cut a representative $BiVO_4$ particle embedded in the LMP metal matrix into a slice using the focused ion beam (FIB) technique. The sectional view of the slice of a single $BiVO_4$ particle together with the LMP metal reveals an intimate interface contact between the faceted $BiVO_4$ and the LMP metal matrix (Supplementary Fig. S6).

To verify our hypothesis that the bioinspired film has advantageous performance for photocarrier collection, control samples of $BiVO_4$ particles deposited on the surface of fluorine-doped tin oxide (FTO) and onto LMP metal film substrates were prepared by conventional methods (Supplementary Fig. S7). To evaluate the effect of the contacting configuration with the substrate on charge collection, surface photovoltage microscopy (SPVM)[44] was employed to map the photogenerated charge distribution on the $BiVO_4$ surface for particles

deposited onto the different substrates and embedded into the substrates (Fig. 2). SPVM images of all $BiVO_4$ particles show positive SPV signals, implying that photogenerated holes are separated to the $BiVO_4$ surface (Fig. 2a–c). However, the spatial distribution and the magnitude of positive SPV signals differ for them (Fig. 2d–f). For $BiVO_4$ particles deposited onto the FTO substrate (Supplementary Fig. S8a), the SPV signals of {011} facets are approximately 60 mV and are higher than that of {010} facet by 40 mV (Fig. 2d). The anisotropic SPV signals are attributed to a larger upward band bending for $BiVO_4$ {011} facets[45]. Therefore, the charge separation within the $BiVO_4$ particles dominates the surface hole collection. When changing the FTO substrate with the LMP metal film substrate (Supplementary Fig. S8b), both SPV signals for {011} facets and for the {010} facet increase by 20 mV (Fig. 2e), indicating that more holes are collected at the $BiVO_4$ surface owing to an additional charge separation process between the $BiVO_4$ particles and the LMP metal film substrate. After embedding the $BiVO_4$ particles into the LMP metal film to facilitate their contact (Supplementary Fig. S8c), the positive SPV signals on the $BiVO_4$ surface are further enhanced by the effective charge separation between them (Fig. 2f). More strikingly, the enhanced SPV signals on {010} facets exceed the signals on {011} facets, indicating that the process is completely dominated by the charge separation between the $BiVO_4$ particles and the substrates with holes collected at $BiVO_4$ surface and electrons collected within the substrates. Therefore, we conclude that the embedding-type interface is responsible for the high-performance photocarrier collection by facilitating the efficient charge separation between the $BiVO_4$ particles and the substrates.

Photochemical deposition provides a visualized identification to track the location of photocarriers. The facet-aided photocarrier separation has been confirmed by the deposition of oxidation- and reduction- products according to previous reports[43,46]. In line with the KPFM results, when $BiVO_4$ particles are simply deposited on the surfaces of FTO and LMP metal films, photoreduction deposited $MnO_x$ nanosheets ($MnO_4^- + e^- \rightarrow MnO_x$) occur dominantly on the {010} facets of $BiVO_4$ particles rather than on the FTO or the LMP metal charge collectors (Fig. 2g, h), indicating that photogenerated electrons are not properly collected by the FTO (or the LMP metal film) substrate (Fig. 2j, k). In contrast, the $MnO_x$ nanosheets are mainly located at the metal charge collector when $BiVO_4$ particles are embedded in the LMP metal (Fig. 2i), indicating more efficient electron collection by the metal substrate (Fig. 2l). The photochemical deposition results unequivocally prove that the bioinspired $BiVO_4$ photoactive film with embedding-type interface has much superior photocarrier collection ability to the those without embedding-type interface.

## Scalable fabrication of the robust, efficient bioinspired photoactive film

As the synthesized $BiVO_4$ sample has a particle size of a micrometer scale, which is far larger than the electron diffusion length (<100 nm) in it[47,48], the photogenerated charge carriers in bulk can be barely collected by the conductive substrate. Accordingly, the FTO-based control photoelectrode (FTO/$BiVO_4$) without embedding-type interface delivered negligible photocurrent density (a few $\mu A\,cm^{-2}$) under AM 1.5 G simulated sunlight irradiation (Supplementary Fig. S9). Surprisingly, the bioinspired $BiVO_4$ photoactive film assembled on quartz using the Field's metal (Metal−$BiVO_4$) produced a photocurrent density of ~ 1 mA cm$^{-2}$ at potential of 1.23 $V_{RHE}$ (vs. reversible hydrogen electrode/RHE) (Supplementary Fig. S9), which is the highest value achieved on undoped micrometer-sized $BiVO_4$ photoelectrodes. The remarkable improvement (>260 times) in the photocurrent density confirms the advantages of the embedding structure of LMP metal and semiconductor particles in enhancing the photocarrier collection efficiency (Fig. 3a). This improvement can be attributed to the greatly enlarged charge carrier collection interface that shortens the charge

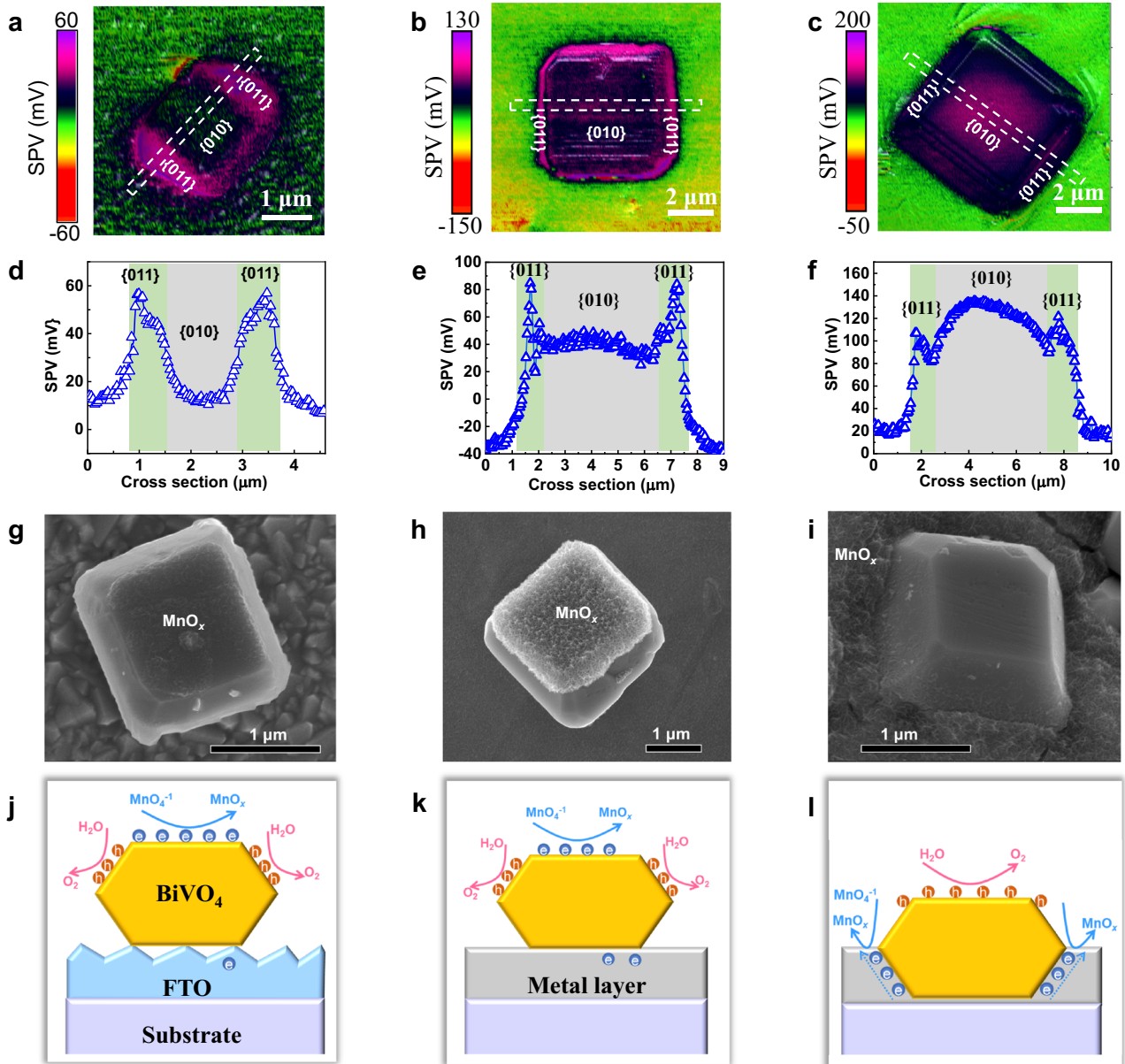

**Fig. 2 | Photogenerated carrier regulation in the bioinspired photoactive film.** **a–c** SPVM images of typical BiVO$_4$ particles with well-developed facets assembled on FTO, on the surface of LMP metal film, and embedded in the metal film, respectively. **d–f** SPV curves plotted along the lines crossing over {011} and {010} facets in (**a–c**), respectively. **g–i** Photo-reduction deposition of MnO$_x$ (MnO$_4^-$ + e$^-$ → MnO$_x$) on BiVO$_4$ particles with well-developed facets assembled on FTO, on the surface of LMP metal film and embedded in the LMP metal film, respectively. **j–l** Sketch maps of spatial separation of photogenerated carriers (electrons and holes) in the BiVO$_4$ particles with well-developed facets assembled on FTO, on the surface of LMP metal film, and embedded in the metal film, respectively.

carrier diffusion lengths needed as depicted in Fig. 1b. As further verification to this elucidation, the control electrode of BiVO$_4$ deposited on the surface of LMP metal film (Metal/BiVO$_4$) without embedding-type interface was also evaluated. Likewise, the particles-deposited photoelectrode delivers a considerably smaller (one-eighth of the later) photocurrent than the bioinspired one (Fig. 3a and Supplementary Fig. S9), again confirming that the performance enhancement dominantly origins from the enlarged strong contact interface rather than the different properties of substrates. Moreover, the bioinspired photoelectrode delivers considerably higher (>2 times) photocurrent densities than the control photoelectrode of faceted BiVO$_4$ particles with similar morphology grown on FTO (FTO@BiVO$_4$) (Fig. 3a and Supplementary Fig. S10) substrate prepared by a hydrothermal process (Supplementary Fig. S9), and clearly rivals with those of BiVO$_4$

photoelectrodes with similar morphology grown on FTO substrates reported (Supplementary Table S1). When an efficient oxygen evolution catalyst (OEC), CoBi, was decorated onto the photoanode, the activity of bioinspired BiVO$_4$ film was further enhanced, reaching a photocurrent density of 1.2 mA cm$^{-2}$ and a low photocurrent onset potential of around 0.2 V$_{RHE}$ (Supplementary Fig. S11a). The performance is almost identical to that of the bioinspired BiVO$_4$ film measured in the presence of hole sacrificial agents (SO$_3^{2-}$), showing a maximum power point (MPP) of around 0.7 V$_{RHE}$ with the applied bias photon-to-current conversion efficiency (ABPE) of 0.41% (Supplementary Fig. S11b).

Besides activity, high stability is equally important for the industrial application of photoelectrodes in solar to hydrogen production. After CoBi OER cocatalyst decoration, the photocurrent density was

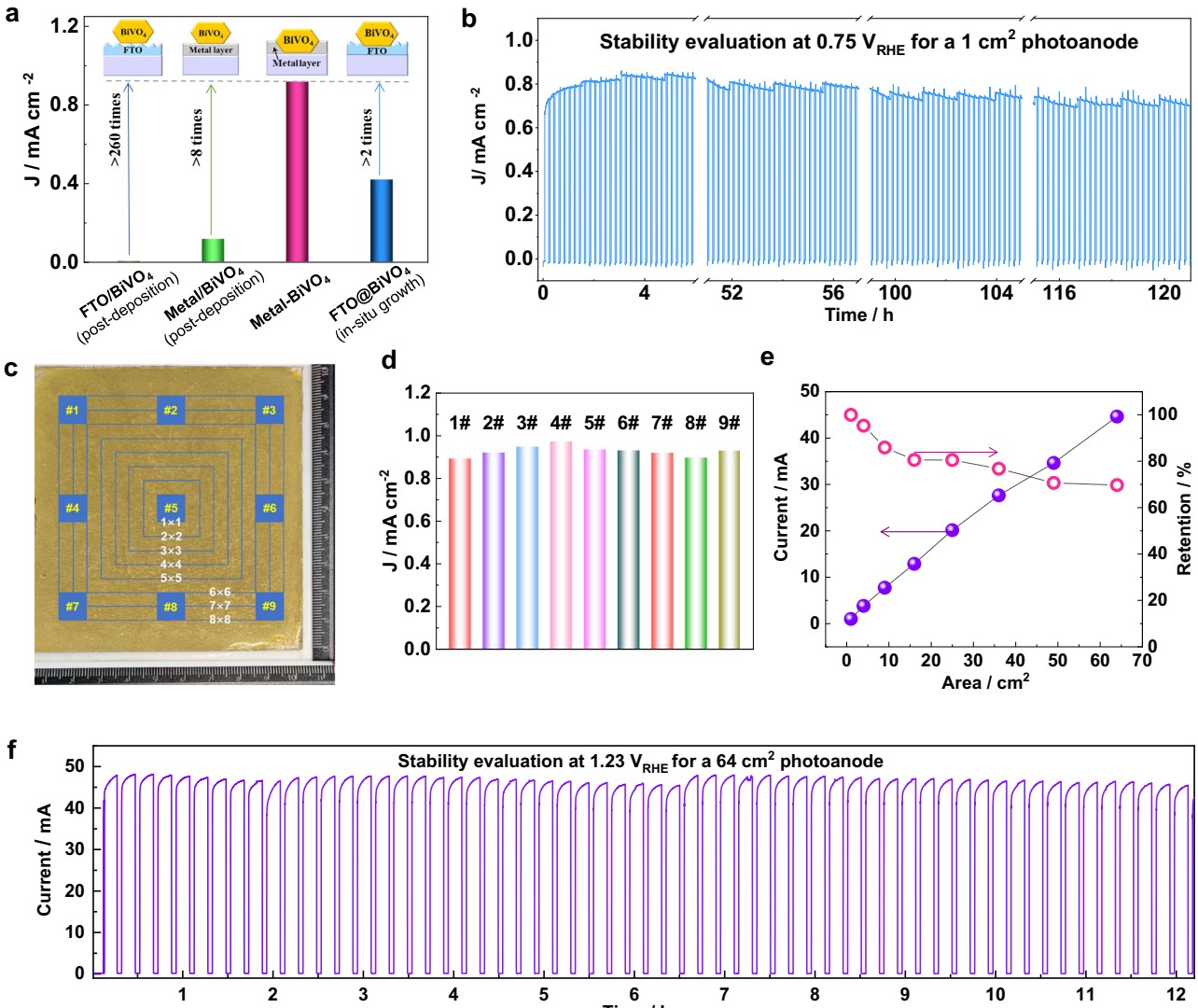

**Fig. 3 | Performance of BiVO₄ particles embedded photoelectrode. a** The comparison of photocurrent density of the photoelectrodes of LMP metal embraced BiVO₄ particles (Metal−BiVO₄), LMP metal supported BiVO₄ particles (Metal/BiVO₄), the FTO-supported BiVO₄ particles (FTO/BiVO₄), and BiVO₄ particles in situ grown on FTO (FTO@BiVO₄) at 1.23 $V_{RHE}$. **b** The long-term stability of the BiVO₄ particle-embedded photoelectrode film with CoBi surface modification at a potential of 0.7 $V_{RHE}$. **c** The optical image of the BiVO₄ particle-embedded photoelectrode with a large size of 10 × 10 cm², in which nine local zones (1 × 1 cm² in area) at the center (#5), edges (#2, 4, 6, 8) and corners (#1, 3, 7, 9) were selected for the photo activity measurements, and different areas (1 × 1, 2 × 2, 3 × 3, 4 × 4, 5 × 5, 6 × 6, 7 × 7 and 8 × 8 cm²) at the center were chosen for the assessment of "area effect". **d** The photocurrent densities of the nine local zones in the photoelectrode after decorating a CoBi cocatalyst at 1.23 $V_{RHE}$. **e** The photocurrents recorded on different areas of the BiVO₄ particle-embedded photoelectrode and the corresponding photocurrent density retentions of operation zones with different areas at 1.23 $V_{RHE}$. **f** The photocurrent decay curves were recorded from the 64 cm² operation area at 1.23 $V_{RHE}$.

evaluated under long-term operation conditions at the MPP (0.7 $V_{RHE}$) as a stability test of the bioinspired BiVO₄ photoanode film. Strikingly, the photoanode maintains its photocurrent density for over 120 h (Fig. 3b), demonstrating reliable performance and stability. As photogenerated holes accumulate on the BiVO₄ particle surface upon light irradiation, photoelectrochemically deposited CoBi majorly distributes on the BiVO₄ particles. However, the trace of CoBi has also been recognized on the Field's metal at the 1.23$V_{RHE}$ deposition potential, as confirmed from the SEM images and corresponding EDS mappings (Supplementary Fig. S12). So, the deposited CoBi might passivate the Field's metal surface together with the pre-formed oxide layer from further oxidation, enabling the long-term stability of the synthesized electrodes. During the stability testing, the oxygen bubbles were clearly observed on the surface of the photoelectrode film (Supplementary Fig. S13), confirming the water-to-oxygen reactions.

The polar $J$–$V$ curves almost overlapped before and after the long-period stability test (Supplementary Fig. S14), further confirming its long-term stability. This is probably attributed to the robust embedding microstructures of the bioinspired film, evidenced by the similar morphology before and after the stability test (Supplementary Fig. S15). These results show that the LMP metal-based PiP strategy is highly effective in achieving high photoelectrochemical performance and stable operation.

The uniformity of large-area photoelectrode film is of critical importance for industrial-scale applications. The photoelectrode performance is usually negatively correlated with the film area, which is often termed the "area effect". The area effect has been frequently reported on BiVO₄ photoanodes[27–29], whose photocurrent density drops by more than 60% when the electrode area increases from 1 to 50 cm⁻². How to eliminate or mitigate the "area effect" remains a

challenge for the development of industrial-scale PEC devices for practical water-splitting reactions.

As the LMP metal-based photoelectrode can be uniformly fabricated in a facile and scalable PiP process as depicted in the above section, we assembled a large area ($10 \times 10$ cm$^2$) BiVO$_4$ particle-embedded LMP metal film on a quartz substrate to assess its activity uniformity (Fig. 3c). The polar $J$–$V$ curves of PEC water oxidation at nine different local zones with area of $1 \times 1$ cm$^2$, including the center (#5), edges (#2, 4, 6, 8) and corners (#1, 3, 7, 9), show very close photocurrent densities at a given potential (Supplementary Fig. S16), confirming the high activity uniformity of the large area film. With the aid of a CoBi co-catalyst, PEC performance at the nine zones can be further enhanced (Supplementary Fig. S17) with short-circuit photocurrent densities ($J_{sc}$) of ~ 1 mA cm$^{-2}$ (Fig. 3d). Meanwhile, we monitored the photocurrent changes as a function of irradiation area (1, 4, 9, 16, 25, 36, 49, and 64 cm$^2$) at the central part (Supplementary Fig. S18). The short-circuit photocurrent ($I_{sc}$) at 1.23 V$_{RHE}$ shows almost a linear increase with the irradiation area, approaching ~45 mA when the operation area reaches 64 cm$^2$. The normalized photocurrent density can retain almost 70% of the value recorded from the unit area when the irradiation area was enlarged up to 64 cm$^2$ (Fig. 3e). Over an irradiation timespan of over 12 h, the photoelectrode well maintained its photocurrent (Fig. 3f), accompanied by uniform bubbles releasing from the surface (Supplementary Fig. S19). In comparison, the photocurrent density of the control faceted BiVO$_4$ photoanode grown on FTO (FTO/BiVO$_4$-G) only retains less than 40% of the value collected from the unit area when the irradiation area was increased to 49 cm$^2$ (Supplementary Fig. S20). From the photoelectrochemical impedance analysis (Supplementary Fig. S21), the LMP metal embraced BiVO$_4$ photoelectrode (Metal–BiVO$_4$) has a much smaller charge transfer resistance ($R_{ct}$) than that of FTO/BiVO$_4$-G, which is in accordance with their PEC performance change trend. The most important is the system series resistance of Metal–BiVO$_4$ is only 0.76$\Omega$, over one order of magnitude smaller than that (9.86$\Omega$) of FTO/BiVO$_4$-G. Therefore, the Ohmic loss is remarkably reduced with increasing the operating area for Metal–BiVO$_4$ as compared with FTO/BiVO$_4$-G, leading to a much better photocurrent density retention for Metal–BiVO$_4$. These results well prove that the LMP metal-based PiP technique endows the photoelectrode with very high activity uniformity for scalable PEC cells and effectively overcomes the "area effect". Note that due to the limited lens size of our solar simulator, the maximum area of the photoelectrode that can be illuminated is 64 cm$^2$.

## Potentiality of the PiP technique in artificial photosynthesis

As discussed above, we have originally developed an LMP metal-based PiP technique to produce a bioinspired photoelectrode film with robustness and scalability for efficient PEC water oxidation. To further unfold the potential applications of the PiP technique in artificial photosynthesis, we further attempted to use it to embed two different types of semiconductor particles in the LMP liquid metal simultaneously to produce a bioinspired photoactive film for artificial photosynthesis, in which two types of embedded semiconductors are bridged with the LMP liquid metal matrix and perform as the HER photocatalyst (HERP) and OER photocatalyst (OERP), respectively (Fig. 4a, b).

The effectiveness of combining p-type Rh doped SrTiO$_3$ (Rh:SrTiO$_3$) as the HER photocatalyst with the n-type BiVO$_4$ as the OER photocatalyst in the solid Z-scheme system with gold film or indium tin oxide particles as a conductive mediator for photocatalytic overall water splitting has been well demonstrated in the pioneering studies[11,12]. To evaluate the potential advantages of liquid metal in constructing a solid Z-scheme photocatalytic system, we chose Rh: SrTiO$_3$ (Supplementary Fig. S22) and faceted BiVO$_4$ as model materials. Prior to the integration, the Rh: SrTiO$_3$ and faceted BiVO$_4$ particles were first decorated with Ru/CrO$_x$ core/shell

cocatalyst for HER and CoO$_x$ cocatalyst for OER, respectively. The decorated particles were then fully mixed and embedded in the liquid metal matrix using the PiP technique to produce a solid Z-scheme system for photocatalytic water splitting (Fig. 4c). The top-viewed SEM image clearly shows that micrometer-sized faceted BiVO$_4$ and sub-micrometer sized Rh: SrTiO$_3$ particles were well embedded in the solidified metal film (Fig. 4d), affirming the effectiveness of the PiP technique in constructing the photoactive film of two semiconductors for artificial photosynthesis. EDS mapping images show the locations of BiVO$_4$ and Rh: SrTiO$_3$ particles, further verifying the effective integration of BiVO$_4$ and Rh: SrTiO$_3$ particles in the Field's metal film according to the spatially complementary distribution between V and Sr elements (Fig. 4e, f).

Mott-Schottky curves and Tauc plots (derived from UV–visible absorption spectra) of Rh:SrTiO$_3$ and faceted BiVO$_4$ were recorded to obtain their flat band potential (approximate Fermi level positions) of 1.9 and 0.2 V$_{RHE}$, and intrinsic band gap of 3.2 eV (with visible light absorption from the impurity level) and 2.3 eV, respectively (Supplementary Fig. S23). Given the energy difference of 4.44 eV between the vacuum level and hydrogen electrode potential[49], the Fermi level of Rh:SrTiO$_3$ and BiVO$_4$ is deduced to be −6.34 and −4.64 eV, respectively. In general, the Fermi level of the electronic conductor should be located between those of the two semiconductors in order to facilitate the Z-scheme charge transfer through their interfaces. Therefore, gold with a Fermi level of −5.1 eV, carbon with a Fermi level from −4.8 to −5.0 eV, and indium tin oxide with a Fermi level of −4.75 eV have been frequently used as the conductor mediator to bridge Rh:SrTiO$_3$ and BiVO$_4$ for constructing Z-scheme system[11–13]. Coincidentally, the Fermi level of Field's metal was evaluated to be around −4.68 eV by UV photoelectron spectroscopy (UPS) (Supplementary Fig. S24), which satisfies the requirement of constructing a Z-scheme system with Rh:SrTiO$_3$ and BiVO$_4$ (Fig. 4g).

To verify the effective Z-scheme charge transfer between the particles of Rh: SrTiO$_3$ and BiVO$_4$ embedded in the LMP metal film, both SPVM and photochemical deposition techniques were employed to monitor the distributions of photogenerated electrons and holes in the film. SPVM image shows positive and negative SPV signals on the adjacent particles of BiVO$_4$ and Rh: SrTiO$_3$, indicating that the photogenerated holes and electrons are separately located at the BiVO$_4$ and Rh: SrTiO$_3$ particles, respectively (Supplementary Fig. S25). This can be further supported by the fact that the photoreduction deposition of MnO$_x$ nanosheets occurs dominantly on the surface of Rh: SrTiO$_3$ particles rather than BiVO$_4$ particles (Supplementary Fig. S26a). All these results solidly prove the effective occurrence of Z-scheme charge transfer in the resulting photoactive film with an embedding-type interface. Accordingly, such an integrated system with an area of 10 cm$^2$ can induce spontaneous water splitting with the release of stoichiometric H$_2$ and O$_2$ upon visible light ($\geq$ 420 nm) irradiation. Its photocatalytic water splitting activity is 2.9 times higher than that of the panel system assembled with the sputtered Au film via the particle transfer method (Fig. 4h). Moreover, the embedding structure with 3D high-quality interfacial contact can greatly enhance the adhesion strength between the semiconductor particles and the metal conductive layer, enabling a high robustness against peeling off under purging and even ultra-sonication treatments. As a consequence, the bioinspired Z-scheme photoactive film is capable of delivering stable photocatalytic water splitting without obvious degradation for a cycling test of 110 h (Fig. 4i). The solar to hydrogen (STH) conversion efficiency of the LMP metal embraced and Au film supported Z-scheme systems was evaluated by monitoring hydrogen production under AM 1.5 G sunlight simulator irradiation (Supplementary Fig. S26b), which are calculated to be ~0.041% for the former and only ~0.009% for the latter. All these remarkable results demonstrate the great superiority of the liquid metal-based PiP technique in constructing efficient solid Z-scheme photocatalytic water splitting systems.

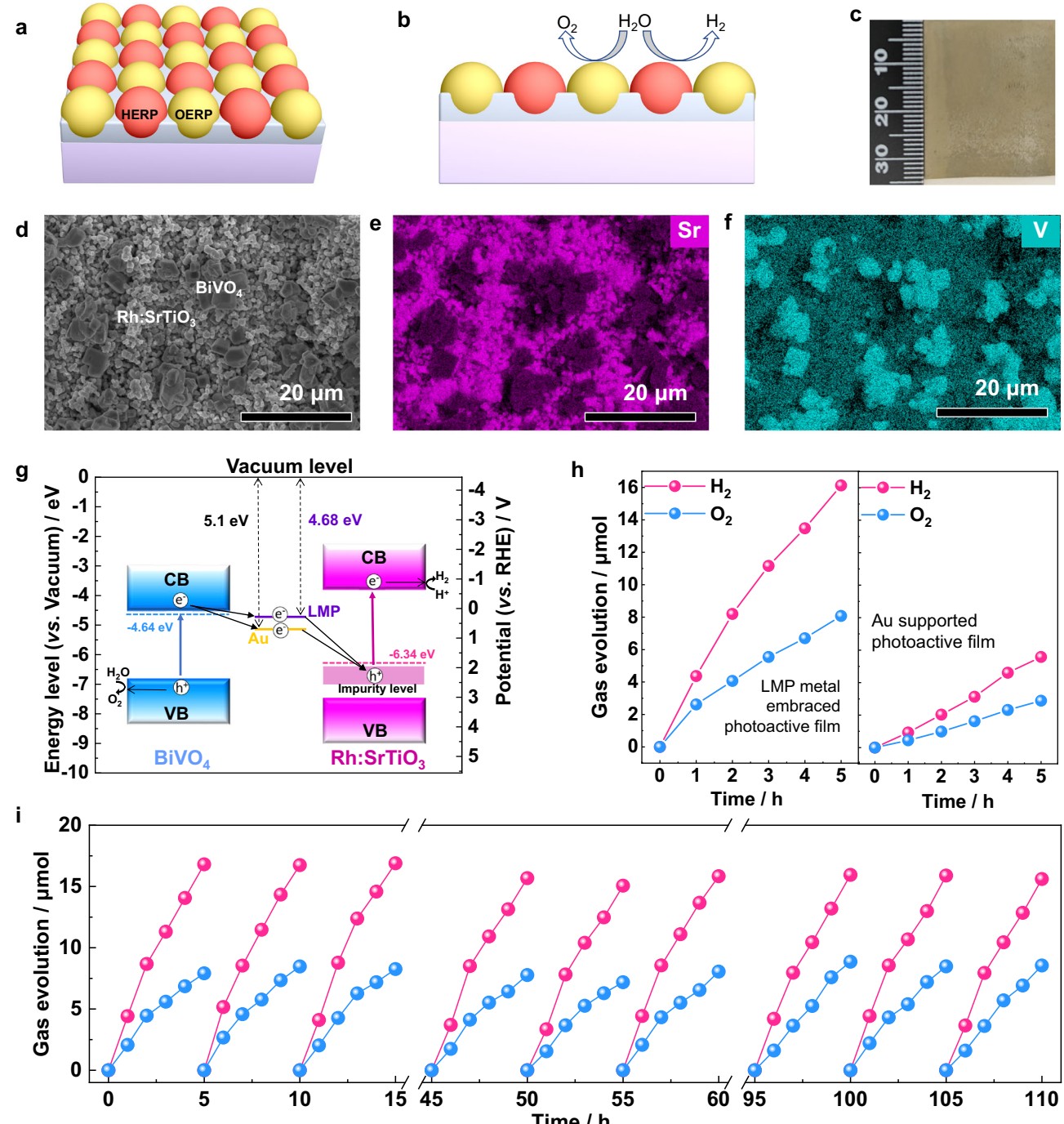

**Fig. 4 | Application of the LMP metal-based PiP process to fabricate Z-scheme photoactive film for artificial photosynthesis. a** Schematic of Z-scheme photoactive film with two types of semiconductors (HERP/OERP) embedded in the LMP liquid metal matrix for artificial photosynthesis. **b** Schematic of water splitting reactions occurring on the Z-scheme photoactive film with two types of semiconductors (HERP/OERP) embedded in the LMP liquid metal matrix. **c** Optical image of the Z-scheme photoactive film comprising of faceted BiVO₄ and Rh:SrTiO₃ particles embedded in the LMP liquid metal matrix. **d** The top-view SEM image of the fabricated Z-scheme photoactive film using the LMP metal-based PiP technique. **e, f** The EDS mapping images of Sr and V recorded on the top-view SEM image. **g** The Fermi energy level positions of different materials. **h** The photocatalytic production of H₂ and O₂ from water splitting with the Z-scheme photoactive films, comprising of Rh:SrTiO₃ and BiVO₄ particles embedded in the LMP liquid metal matrix (left panel) and supported on the Au film (right panel), under visible light (≥420 nm) irradiation. **i** The cycling performance of photocatalytic water splitting conducted with the LMP metal-based Z-scheme photoactive film.

## Universality and sustainability of the bioinspired films by the PiP technique

Various types of semiconductor particles could be assembled on different substrates by the PiP technique because the LMP metal serves as efficient conductive glue between the semiconductor particles and the substrate. The universality of this PiP technique was demonstrated by fabricating different photoelectrode films using commercial semiconductor particles (e.g., ZnO, WO₃, and Cu₂O) with different bandgaps and conductivities. The particle sizes of commercial semiconductor powders are of different scales, from a few hundred nanometers to several micrometers, and these particles tend to aggregate (Supplementary Fig. S27a−c). Besides the particles-

embedded films (Supplementary Fig. S27d–f), the micro-/nanostructured films have also been in situ grown on the FTO substrate for comparison (Supplementary Fig. S27g–i). Surprisingly, the ZnO particles-embedded photoelectrode (Metal–ZnO) delivers much higher photocurrent than its control photoelectrode assembled on FTO (FTO/ZnO), and even the ZnO nanorod array photoelectrode in situ grown on FTO (FTO/ZnO-G) (Supplementary Fig. S28). The $J_{sc}$ (-0.53 mA cm$^{-2}$) at 1.23 $V_{RHE}$ is increased by a factor of 25.4 and 2.9, respectively (Fig. 5a), outperforming most ZnO nanostructured photoelectrodes in situ grown on FTO substrates (Table S2). Similar results are also observed using WO$_3$ and Cu$_2$O particles, in which the photocurrent density is enhanced by a factor of 4.4 and 13 compared with the FTO-based counterparts, respectively (Fig. 5a and Supplementary Fig. S29). A comprehensive comparison with the corresponding photoelectrodes in situ grown on FTO substrates can be found in the supplementary information (Supplementary Fig. S29, Tables S3 and 4), showing the superiority of the particles-embedded photoelectrodes. These results consistently show the universality of the LMP metal-based PiP technique with greatly enhanced PEC performance.

The substrates can also be varied for the LMP-based fabrication of photoelectrodes, including paper, board, cloth, Ti mesh, glass, and Polyethylene terephthalate (PET). Using a commercial semiconductor WO$_3$ as an example, the WO$_3$ particles-embedded photoelectrodes fabricated on different substrates demonstrated very similar PEC performances of over 1 mA cm$^{-2}$ photocurrent density at 1.23 $V_{RHE}$ (Fig. 5b and Supplementary Fig. S30). Therefore, it can be concluded that this LMP metal-based PiP technique can be a generic approach in fabricating photoelectrodes and, extensively, functional thin films.

To demonstrate the scalability of the LMP metal-based PiP technique, we fabricated large-size PET-supported photoelectrode films of 100 mm × 100 mm from commercial semiconductor (ZnO, WO$_3$, and Cu$_2$O) powders. The photoactive film was fabricated by doctor blade printing of the melted LMP metal on a PET substrate, drop casting of semiconductor particles on the surface, and then pressing the particles into the liquid metal film with an industrial roll-to-roll technique (Fig. 5c). The embedded structure with 3D high-quality interfacial contact can enhance adhesion strength between the semiconductor particles and metal conductive layer, giving rise to high robustness against peeling under purging and even ultra-sonication treatments. The large-size flexible photoactive films fabricated can be thus arbitrarily bent without damage owing to the high ductility of the LMP Field's metal as well as the strong adhesion between the WO$_3$ particles and the Field's metal (Supplementary Fig. S31), verifying the scalability and robustness of the LMP metal-based PiP technique.

To further demonstrate this robustness, we conducted a series of bending cycling tests for the flexible WO$_3$ photoelectrode to assess its stability against deformation. Specifically, the particles-embedded photoelectrode film was subject to repeated cycles of bending to a certain curvature and then relaxing back to flat (inset image in Fig. 5d). By monitoring the photocurrent densities of the photoelectrode film after a large number of bending cycles (Supplementary Fig. S32), a current density retention curve along with cycle numbers was recorded (Fig. 5d). As anticipated, the embedded photoelectrode structure with LMP metal possesses strong stability against cyclic bending as the photocurrent density can retain over 95% of the initial value even after 10$^5$ bending cycles. This strong stability warrants promising applications in other fields, such as in portable optoelectronic devices.

The cost of photoactive films for water splitting in practical applications is an important factor to consider. Although the use of LMP metal (Field's metal, Bi: In: Sn = 32.5: 51: 16.5 wt%) contains scare In metal, this PiP technique has the feature of high material recyclability just by applying ultra-sonication treatment in hot water to separate the semiconductor particles from the LMP (Field's) metal. For example, by ultra-sonicating the LMP metal embraced photoactive film in hot water of >62 °C, Field's metal melts to release all active materials (Fig. 5e and

Supplementary Fig. S33). We further fabricated particles-embedded photoactive films using the recycled Field's metal. The resulting photoactive films deliver almost identical photocatalytic overall water-splitting activity and PEC performance (Fig. 5f and Supplementary Fig. S34), confirming the high recyclability of the Field's metal. The wide choice of semiconductor particles and substrate materials, as well as the recyclability of these LMP materials, promises a low-cost and sustainable manufacturing process.

In this study, we demonstrated the proof-of-concept for producing bioinspired photoactive films for photoelectrochemical or photocatalytic reactions by a developed PiP technique of embedding semiconductor particles in LMP liquid metal (e.g., Field's metal) as a conductive matrix. The constructed photoelectrode film with a strong 3D semiconductor/metal interface has a high photocarrier collection ability and activity uniformity. As a model system, the BiVO$_4$ particles embedded in LMP liquid metal film deliver 260 times (or 8 times) higher photoanodic current density in photoelectrochemical water oxidation than that assembled on FTO (or LMP liquid metal film) without embedding-type interface and can be stably operated over 120 h. Such a film realizes the remarkable record of -70% of its initial performance when scaled from 1 to 64 cm$^2$. Furthermore, the photocatalytic film of Rh: SrTiO$_3$ and BiVO$_4$ particles-embedded LMP liquid metal can realize the very stable spontaneous water splitting with the release of stoichiometric H$_2$ and O$_2$ under visible light irradiation. It is anticipated that the performance of photoactive films could be further improved by matching the work functions of the LMP metal and semiconductors to form an Ohmic contact. Coincidentally, an added benefit of such films is that a wide range of LMP metals and semiconductors can be used and retrieved, providing sufficient room for the construction of more efficient photoelectrodes and Z-scheme photocatalytic sheets by properly adjusting the work function of LMP metals according to the Fermi levels of semiconductors. All in all, this LMP metal-based PiP technique promises a low-cost scalable processing route for solar energy conversion devices and applications.

## Methods
### Materials
Bi(NO$_3$)$_3$·5H$_2$O(≥99.0%), NH$_4$VO$_3$(≥99.0%), NaCl(≥ 99.8%), NH$_3$·H$_2$O (~28%), HNO$_3$(~68%), HBO$_3$(≥99.8%), KOH(≥85.0%), Na$_2$SO$_3$(≥ 97.0%), Co(NO$_3$)$_2$·6H$_2$O (≥98.5%), KMnO$_4$(≥ 99.5%), SrCO$_3$(≥99%), TiO$_2$(≥99%), Rh$_2$O$_3$(≥ 99.8%), RuCl$_3$·nH$_2$O(≥98%), K$_2$CrO$_4$(≥98%), polyvinyl alcohol(≥99%), Zn(NO$_3$)$_2$·6H$_2$O(≥99%), Na$_2$WO$_4$·2H$_2$O(≥99.5%), HCl (36.0-38.0%), (NH$_4$)$_2$C$_2$O$_4$·H$_2$O(≥99.8%), CuSO$_4$·5H$_2$O(≥99%), lactic acid, NaOH(≥96%), and TiCl$_4$(≥98.0%) were purchased from Sinopharm Chemical Reagent Co., Ltd. ZnO (99.9%), Cu$_2$O(99.9%), WO$_3$ (≥99.0%) and low-melting-point liquid metal (bismuth indium tin ingot/Field's metal, Bi: In: Sn = 32.5: 51: 16.5 wt%) is purchased from Thermo Fisher Scientific Inc. All chemicals were used without further purification.

### Synthesis and modification of BiVO$_4$ and Rh doped SrTiO$_3$ (Rh:SrTiO$_3$) powders
The BVO$_4$ particles were synthesized by a hydrothermal method. In brief, Bi(NO$_3$)$_3$·5H$_2$O (9 mmol) and NH$_4$VO$_3$ (9 mmol) were dissolved in 55 mL and 20 mL aqueous nitric acid solution (2.0 M), respectively. After mixing these two solutions, ammonia aqueous solution (11 M) was then added to adjust the pH around 2. The solution was magnetically stirred to generate orange precipitates after which 55 mg NaCl was added under stirring. After aging for 2 h, the so-formed suspensions were transferred into a Teflon-lined autoclave and were heated at 200 °C for 24 h. The yellow powdery product was separated by filtration, rinsed with deionized water, and dried at 60 °C in air. The Rh:SrTiO$_3$ particles were synthesized by a solid-state reaction. SrCO$_3$, rutile TiO$_2$, and Rh$_2$O$_3$ powders were mixed in a mortar at a Sr/Ti/Rh ratio of 1.05:0.97:0.03 and ground for around 1 h. The mixture was

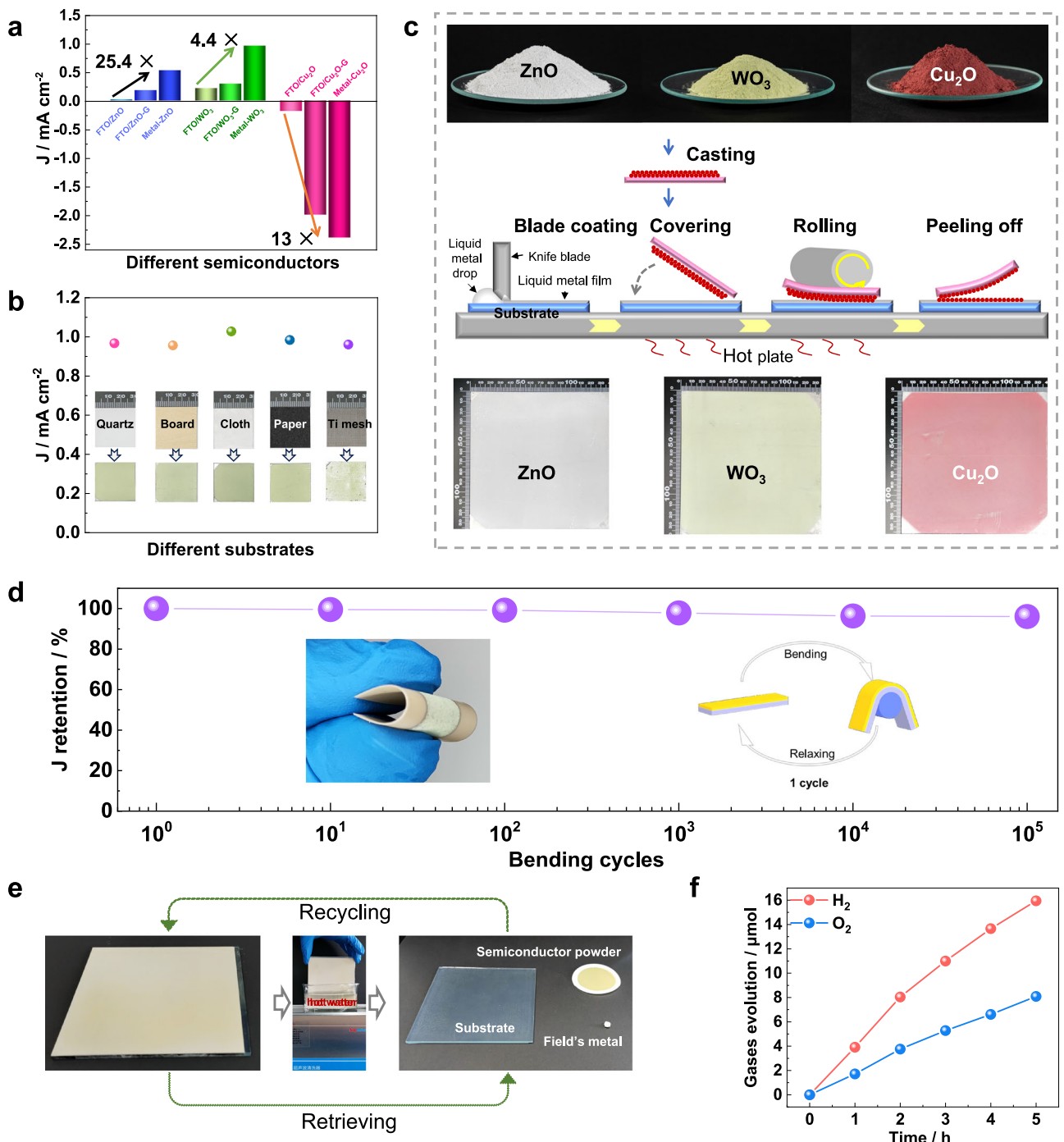

**Fig. 5 | Advantages of the LMP metal-based PiP technique for the synthesis of photoactive films. a** The short-circuit photocurrent density enhancement of the photoelectrodes (Metal–ZnO, Metal–WO₃, Metal–Cu₂O) of LMP metal embraced particles relative to the control photoelectrodes (FTO/ZnO, FTO/WO₃, FTO/Cu₂O) assembled on FTO for commercial ZnO, WO₃, and Cu₂O powders and the micro-/nanostructured film photoelectrodes ((FTO/ZnO-G, FTO/WO₃-G, FTO/Cu₂O-G; G from grown) of ZnO, WO₃, and Cu₂O in situ grown on the FTO substrate. **b** The short-circuit photocurrent densities of the particles-embedded photoelectrodes from commercial WO₃ powder on different substrates (quartz, board, cloth, paper, and Ti mesh). **c** Schematic of scalable fabrication of the LMP metal embraced photoactive films on PET from commercial semiconductor (ZnO, WO₃, and Cu₂O) powders using an industrial roll-to-roll process. **d** The PEC performance of the WO₃ particles-embedded photoelectrode film on PET for the cyclic bending tests. The inset is the optical image of the film photoelectrode assembled on a PET substrate under bending conditions and the schematical image of a bending cycle of the photoelectrode. **e** The optical images of a 100 mm × 100 mm film of BiVO₄ and Rh:SrTiO₃ semiconductor particles embedded in Field's metal on a glass substrate and the raw materials retrieved from the film by ultrasonication treatment in a hot water bath. **f** The photocatalytic overall water splitting activity of BiVO₄ and Rh:SrTiO₃ particles-embedded photoactive film assembled with the retrieved Field's metal under visible light (≥420 nm) irradiation.

calcined at 1100 °C for 10 h in a muffle furnace to produce highly crystallized Rh:SrTiO₃ powder. The CoOₓ (0.5 wt%) was loaded on the BiVO₄ by photo-deposition from a solution of Co(NO₃)₂·6H₂O. A Ru (0.5 wt%)/Cr₂O₃ (0.5 wt%) core/shell cocatalyst was loaded on Rh:SrTiO₃ via a two-step photo-deposition from the solution containing RuCl₃·nH₂O and K₂CrO₄, sequentially, in vacuum.

## Preparation of particles-embedded photoactive films by the PiP technique

In a typical fabrication, the Field's metal was heated above its melting point (62 °C) and was then coated onto the substrate ($1 \times 3$ cm$^2$) to form a metal film with a knife blade. Meanwhile, 20 mg of BiVO$_4$ photocatalyst particles dispersed in 2 mL of isopropanol were dropped onto the liquid metal film to assemble the photocatalyst layers. Subsequently, the photocatalyst particles were pressed into the liquid metal film by a silicone rubber roller when the metal was molten. Those particles that were loosely attached to the metal film were blown off by the nitrogen flow when the liquid metal was solidified at room temperature. Pre-heating this LMP metal embraced photoanodes at 200 °C in the air was conducted to form a thin oxide layer atop the metal film to protect the metal film from being further oxidized during PEC or photocatalytic applications.

The large-size photoactive film was readily prepared by the same procedures just increasing the substrate size and suspension volume. For electrical connections, a copper wire was fused with the liquid metal by soldering and was sealed by melting adhesive. The bioinspired artificial photosynthesis film was readily prepared by the same procedures just using the mixture particles of BiVO$_4$ and Rh:SrTiO$_3$ with a mass ratio of 2:3.

## Preparation of the control photoactive films assembled on the surface of FTO

Fluorine-doped tin oxide (FTO) glass was washed sequentially in ethanol, acetone, and isopropanol under ultrasonication each for 20 min and was dried in nitrogen gas flow before use. In total, 20 mg of photocatalyst particles were ultrasonically dispersed in 2 mL of isopropanol to form a suspension. The suspension was then dropped onto the FTO ($1 \times 3$ cm$^2$) substrate to assemble the photocatalyst layers. Subsequently, 10 μL of TiCl$_4$ ethanol solutions (10 mM) was dropped onto the photocatalyst layers and dried in air. This procedure was repeated three times. The FTO-based photoelectrode was then heated at 300 °C for 2 h in air (or Ar for materials that are vulnerable to being heated in air) as the conventional particle-post-assembly technique.

## Preparation of the control photoactive film assembled on the surface of LMP metal

The LMP metal (Field's metal) was heated above its melting point (62 °C) and was coated on the substrate ($1 \times 3$ cm$^2$) to form a liquid metal film with a knife blade. In total, 20 mg of photocatalyst particles were ultrasonically dispersed in 2 mL of isopropanol to form a suspension. The suspension was dropped onto the LMP metal film to assemble photocatalyst layers. The LMP metal-based control photoelectrode was obtained by heating to melt and then cooling the substrate to room temperature.

## Preparation of the control photoactive films in situ grown on FTO glass

The BiVO$_4$ photoelectrode in situ grown on FTO was fabricated by a seed-assistant hydrothermal method. The BiVO$_4$ seed layer was firstly cast on the FTO substrate: 0.6468 g of Bi(NO$_3$)$_3$·5H$_2$O was dissolved in 2 mL of 68% HNO$_3$ and followed by adding 4 mL H$_2$O and 0.334 g of polyvinyl alcohol (PVA) under stirring for 12 h to dissolve PVA in this aqueous solution. Then, 0.156 g of NH$_4$VO$_3$ was added to the solution, and its color changed from orange to blue. The 200 μL blue solution was then dropped and spin-coated on cleaned FTO substrates at 1000 rpm for 60 s, followed by a calcination at 450 °C in air for 2 h. For the hydrothermal growth of BiVO$_4$ photoelectrode: 0.1164 g of Bi(NO$_3$)$_3$·5H$_2$O and 0.028 g of NH$_4$VO$_3$ were dissolved in 1.6 mL of 68% HNO$_3$ and followed by adding H$_2$O up to 60 mL. In total, 20 mL of the resulting solution was transferred to a Teflon-lined autoclave. The FTO substrate with the seed layer was immersed in the solution with the conducting side facing down. The sealed autoclave was heated at 180 °C for 12 h. The BiVO$_4$ photoanode was then obtained after calcination at 450 °C in the air for 2 h.

The ZnO photoelectrode in situ grown on FTO (FTO/ZnO-G) was prepared by electrochemical deposition. 0.76 g of zinc nitrate hexahydrate (Zn(NO$_3$)$_2$·6H$_2$O) was dissolved in 100 ml H$_2$O as the electrolyte. The temperature of the electrolyte was maintained at 75 °C in a water bath. The ZnO nanorod films were grown at a constant voltage of 2.5 V for 30 min using a direct-current power supply in a two-electrode configuration (a Pt foil as the counter electrode). FTO/ZnO-G photoanode was obtained by calcined in air at 500 °C for 2 h after a clean and dry process.

The WO$_3$ photoelectrode in situ grown on FTO (FTO/WO$_3$-G) was synthesized by a hydrothermal method. 0.132 g of sodium tungsten dehydrate (Na$_2$WO$_4$·2H$_2$O) was dissolved in 17.2 mL of H$_2$O under stirring. Then, 5.7 mL of 3 M HCl was added drop by drop to form a yellowish precipitate, followed by the addition of 0.131 g of ammonium oxalate monohydrate ((NH$_4$)$_2$C$_2$O$_4$·H$_2$O) into the above suspension. Then deionized water was added to get a total volume of 40 mL solution. The as-prepared precursor was then transferred into a Teflon-lined stainless autoclave. Two pieces of FTO ($1 \times 3$ cm$^2$) substrate were immersed and leaned against the wall of the Teflon vessel with the conducting side facing down. After the autoclave was sealed, the hydrothermal synthesis was carried out at 120 °C for 12 h. After that, the autoclave was allowed to cool down to room temperature in the oven; then, the FTO substrates were taken out and rinsed with deionized water several times and dried at 60 °C in ambient air. The as-prepared films were calcined in air at 450 °C for 1 h.

The Cu$_2$O photoelectrode in situ grown on FTO (FTO/Cu$_2$O-G) was prepared by electrochemical deposition. 5 g of copper (II) sulfate pentahydrate (CuSO$_4$·5H$_2$O) was dissolved in a 100 ml mixture of aqueous solution that contained 22.4 mL of lactic acid solution. The bath pH was adjusted by adding 16 g NaOH, and the temperature of the solution was maintained at 30 °C in a water bath. The Cu$_2$O films were grown at a constant current density of -0.1 mA cm$^{-2}$ for 220 min using a direct-current power supply in a two-electrode configuration (a Pt foil as the counter electrode). FTO/Cu$_2$O-G photocathode was obtained after a clean and dry process.

## Preparation of Au-supported photoactive film by the particle transfer method

In a typical fabrication according to the previous report (ref. 36). 24mg of Rh:SrTiO$_3$ with Ru/CrO$_x$ core/shell cocatalyst particles and 16 mg of faceted BiVO$_4$ with CoO$_x$ cocatalyst were dispersed in 5 mL of isopropanol, and the dispersion solution was dropped onto the substrate ($3.3 \times 3$ cm$^2$) to assemble the photocatalyst layers. Subsequently, after drying at room temperature, a 500 nm-thick Au layer was deposited with thermal evaporation equipment (base pressure of about $2 \times 10^{-4}$ Torr and deposition rate of about 1 Å/s). After that, a 2 μm-thick Ti layer was deposited via DC magnetron sputtering (KC-CK350), using metal titanium (purity: 99.995%, size: φ 50.8*3 mm) as the target in argon. The Ti film was sputtered at 100 W under a vacuum chamber pressure of 2 Pa for 60 min. Subsequently, the Ti and Au conductive layers were peeled off together with a layer of particles with the aid of conductive tape. Lastly, the film was fixed on a quartz substrate and blown off the loosely attached particles.

## Photochemical deposition for tracing photogenerated carriers

Photo-reduction deposition of MnO$_x$ from a solution containing MnO$_4^-$ was referred to in a previous report (ref. 46). In brief, the photoelectrodes were immersed into the KMnO$_4$ aqueous solution (1.65 mM). The photoelectrodes were then irradiated by a 300 W Xenon light for 15 min. The redox reactions by photo-generated

charges (i.e., $e^-$ and $h^+$) can be described as the following equations:

$$MnO_4^- + e^- \rightarrow MnO_x \tag{1}$$

$$H_2O + h^+ \rightarrow H^+ + O_2 \tag{2}$$

## Decoration of CoBi on the BiVO₄ photoactive films

The CoBi cocatalyst was decorated on BiVO₄ photoelectrodes via a photo-assisted electrodeposition process. The process was conducted in a traditional three-electrode photoelectrochemical cell (working electrode: BiVO₄ photoelectrode, reference electrode: Ag/AgCl electrode, counter electrode: Pt foil) with the electrolyte of 1.0 M potassium borate buffer aqueous solution containing 0.5 mM $Co(NO_3)_2$ (pH = 9). A potentiostatic deposition was implemented at 1.23 $V_{RHE}$ for 10 s under the irradiation of a 300 W Xe arc lamp coupled with an AM 1.5 G filter. After the deposition, the photoelectrode was rinsed with deionized water.

## Photoelectrochemical measurements

The photoelectrochemical water splitting test was carried out in a three-electrode system with an electrochemical station (Biologic VSP 300), in which the photoelectrode, Ag/AgCl electrode, and the Pt foil were used as the working electrode, reference electrode, and counter electrode, respectively. A potassium borate buffer (1.0 M, pH = 9), which was obtained by adjusting the 1.0 M borate solution pH value to 9 with a certain amount of KOH, is used as the electrolyte. For the test carried out in the presence of hole sacrifice ($SO_3^{2-}$), the electrolyte is potassium borate buffer (1.0 M, pH = 9) containing 0.2 M $Na_2SO_3$. A 300 W Xe arc lamp with an AM 1.5 G filter was used as the light source. The light intensity was calibrated to c.a. 100 mW cm$^{-2}$ with a standard silicon cell detector. The recorded bias vs. Ag/AgCl reference electrode can be converted to potential referring to RHE according to the Nernst equation ($E_{RHE} = E_{Ag/AgCl} + 0.059\ pH + 0.196$). The Mott–Schottky curve was measured between 1.7 $V_{RHE}$ and 2.1 $V_{RHE}$ for Rh:SrTiO₃ and 0 $V_{RHE}$ and 0.6 $V_{RHE}$ for BiVO₄ with an AC amplitude of 10 mV and different frequencies of 0.6, 0.8, 1.0 and 1.2 kHz in a three-electrode system (Ag/AgCl electrode as the reference electrode, Pt foil as the counter electrode, potassium borate buffer (1.0 M, pH = 9) as the electrolyte).

## Photocatalytic overall water-splitting test

The photocatalytic overall water splitting was carried out in an automatic testing system (Perfectlight Sci&Tech, Labsolar-6A). A 300 W Xenon lamp equipped with a cutoff filter ($\lambda \geq 420$ nm) was employed as the light source (Perfectlight Sci&Tech, PLS-SXE-300UV). The sample fixed on a Teflon supporter was soaked in a 250 mL reaction container with 100 mL of deionized water for the water-splitting test under continuous stirring. The generated gas was analyzed by gas chromatography (Shimadzu, GC2014) during the test.

The STH energy conversion efficiency ($\eta$) was calculated according to the following equation:

$$\eta(\%) = \{(R_H \times \Delta G^\circ)/(P \times S)\} \times 100 \tag{3}$$

where $R_H$, $\Delta G^\circ$, $P$, and $S$ denote the rate of H₂ evolution (mol s$^{-1}$) in photocatalytic water splitting, standard Gibbs energy of water ($237.13 \times 10^3$ J mol$^{-1}$), intensity of simulated sunlight (0.1 W cm$^{-2}$) calibrated by a standard silicon cell detector and irradiation area (10.0 cm$^2$), respectively. The light source was an AM 1.5 G solar simulator (PEC-L01, Peccell Technologies, Inc.), and a top-irradiation reaction vessel was used (Perfectlight Sci&Tech, Labsolar-6A). Photocatalytic overall water splitting activity was evaluated in 100 mL of pure deionized water with 10 cm$^2$ Z-scheme photoactive film that

Rh:SrTiO₃ and BiVO₄ particles were embedded in the LMP liquid metal matrix.

## Characterizations

X-ray diffraction patterns of the samples were recorded on a D8 Advance (Bruker, Germany) using Cu $K_\alpha$ irradiation. SEM images and EDS mappings were obtained on a Verios G4 UC (Thermofisher Scientific, America). The work functions of the samples were analyzed by ultraviolet photoelectron spectroscopy (Thermo ESCALAB 250Xi). UV–vis absorption spectra were recorded on a UV-visible-infrared diffuse reflectance spectrophotometer (Jasco V-770).

## Surface photovoltage microscopy (SPVM)

The SPVM was performed under an ambient atmosphere using a commercial AFM system (Bruker Dimension Icon). Platinum/iridium-coated silicon tips with a spring constant of 1–5 N m$^{-1}$ and a resonance frequency of 60–100 kHz (Bruker SCM-PIT) were used. To obtain an SPVM image, surface potential signals were first mapped in the amplitude-modulated (AM-KPFM) mode at an a.c. voltage of 0.5 V and a tip lift height of 50 nm. SPVM measurements were conducted by continuously mapping the surface potential images obtained in the dark and under illumination. The difference between these images obtained at the same location was extracted as an SPVM image. For typical SPVM images, a 420-nm light with a light intensity of approximately 2 mW cm$^{-2}$ was used as the excitation source obtained from a Zolix-monochromator with a 500 W Xe lamp.

## Data availability

The data that support the findings of this study are available from the corresponding author upon request.

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

## Acknowledgements

The authors thank the National Key R&D Program of China (No. 2021YFA1500800), National Natural Science Foundation of China (Nos. 51825204, 52072377, 51972312, and 52188101), CAS Projects for Young Scientists in Basic Research (YSBR-004), Youth Innovation Promotion Association of the Chinese Academy of Sciences (No. 2020192) for the financial support. G. L. thanks the financial support from the New Cornerstone Science Foundation through the XPLORER PRIZE.

## Author contributions

G.L. led the project and H.M.C. guided the project. C.Z. designed experiments, analyzed data, and drafted the paper. X.T.C. performed experiments and collected data. F.T.F. and R.T.C. collected and analyzed SPV data. X.X.X. participated in the data analysis and the paper modification. Y.Y.K. and J.D.G. provided some materials for experiments. L.Z.W., G.Q.L., and K.D. provided critical suggestions and modified the paper. All authors were involved in the data discussion and revising the paper.

## Competing interests

The authors declare no competing interests.
