## [Peer Review File · Nature Communications]

REVIEWER COMMENTS

Reviewer #1 (Remarks to the Author):

This study is a tour de force on scalable photoelectrode manufacturing, which deserves to be published in a high impact journal. The study introduces a particle-implanting technique for scalable photoelectrode and photocatalytic Z-scheme fabrication, which is applied to a range of light absorbers and substrates. The study is thoroughly conducted and will provide value for many researchers in the solar fuels field. However, a few aspects remain to be addressed. The manuscript can be accepted once these points are included in the main text.

1. How do you ensure a homogeneous surface wetting for all substrates employed? Are the pristine Si, FTO or flexible PET substrates used as purchased, or are you performing any surface treatment to improve wetting? Field metal coating can pose challenges due to surface tension, so I am wondering if the $\sim 1\mu\text{m}$ film thickness is homogeneous across all substrates.

2. I don't think the PSI/PSII schematic in Fig. 1a is necessary. The thylakoid lipid membrane operates quite differently from the current approach, and the inorganic oxide particles are quite different from the complex natural photosystems. As mentioned earlier, the study has plenty of merit on its own, without needing this bio-inspired comparison. The same comment applies to the labelling of Fig. 4a, as you are not embedding the natural PSI/PSII complexes in the Field metal, but inorganic oxide particles.

3. Supplementary Fig. 6a is a key plot of the paper and should replace the current Fig 3a. The poor interface of BiVO₄ and FTO without binders is well known, hence, the focus on 260x improvement does not say much. The comparison with in situ, hydrothermally grown FTO|BiVO₄ is much more appropriate and still yields a respectable performance, hence that result should be shown in Fig. 3a as mentioned above.

The authors only describe in the Methods that the particle-post-assembly technique relates to a TiO₂ binder obtained from TiCl₄. The wide bandgap TiO₂ binder may not have the ideal band levels to facilitate charge transfer between other oxides and FTO and may actually act as an insulating barrier. That is why the direct FTO|oxide interface of the in situ grown oxides provides a better comparison.

The concluding paragraph also mentions the "260 times higher photoanodic current density in photoelectrochemical water oxidation than that without embedding-type interface". Again, this refers to a different type of interface (FTO|BiVO₄), whereas the improvement without embedding BiVO₄ powder in the Field metal is only 8x, as shown from the current Fig. 3a.

4. Fig. 3d can be shown as a 2D plot, with sample numbers (1-9) on x-axis. It is hard to compare similar photocurrents on this 3D plot.

5. The 120 h stability of the metal|BiVO₄ photoanode is remarkable. However, Field metal can become oxidized at moderately positive potentials. So, how does the metal|BiVO₄ interface remain stable without decomposing? Does CoBi deposit on the Field metal as well at the 1.23V vs. RHE deposition potential? Supplementary Fig. 11 appears to show a flaky interface; can the authors confirm with EDX if CoBi is passivating the FM surface from oxidation?

6. Does performance depend on surface oxides, or the work function of the low melting alloy? Fig. 4g,h indicates that it could, so more discussion would be welcome.

7. One very general, big picture question is how do the photoelectrodes and photocatalytic Z-schemes behave under heating? How can these systems prevent delamination or degradation under real-world conditions, where solar panels commonly reach over 60C under sunlight irradiation?

This delamination process is presented as recyclability in Fig. 5e, however, the photographs are too small, so the text and details are not really visible. I'd suggest the authors increase the photos, and show larger close-up images in the SI.

8. Fig. 4h: can you mention the STH efficiencies of both Au and Field metal systems? Only the STH of the Field metal system is mentioned later on.

The Methods mention that a $\lambda \geq 420$ nm cutoff filter is used, however, both BiVO₄ and SrTiO₃ are mostly active in the UV region. Is the filter fixed within the solar light simulator? It may be more impressive to show STH values without the $\lambda \geq 420$ nm filter.

9. How does this system compare with similar large scale, flexible Ti mesh or Ti foil photoanodes (10.1002/anie.201207578, 10.1038/s41586-022-04978-6)?

10. Page 21, lines 428/429: the performance of photoelectrodes fabricated on different substrates are not shown in Fig 5a, but they should be added, as this is a key novelty point of the manuscript.

Minor comments:

1. Page 14, lines 274/275: the paragraph suddenly goes from discussing performance to scalability (the scalability discussion continues in the next paragraph). It would be easier to follow if this paragraph would be split into two.

2. Page 31, lines 679/681: "A Ru (0.5 wt%)/Cr₂O₃ (0.5 wt%) core/shell cocatalyst was loaded on Rh:SrTiO₃ via a two-step photo-deposition from the solution containing RuCl₃·nH₂O and K₂CrO₄, sequentially, in vacuum." Is this a solution, or a vacuum process?

3. Page 35, lines 763/764: "a 2 μm-thick Ti layer was deposited via DC magnetron sputtering (KC-CK350), using a metal copper (purity: 99.995%, size: φ 50.8*3 mm) as the target". Is this a Cu or Ti target?

Reviewer #2 (Remarks to the Author):

The authors describe a novel method to embed semiconductor nanoparticles in liquid metal matrixes for photoelectrochemical water splitting. The performance is reported to increase drastically and this is attributed to the strong interaction between particle and metal matrix, which results in efficient charge carrier transport.

As metal matrix, the authors used the Field's metal which is an alloy of Bi, In and Sn. In is particularly expensive. The BiVO₄ particles are prepared by hydrothermal method, which is reported by them in the introduction as not suitable for scaling up. The authors mentioned at the end using recycled materials would be a way around it, but it is not ideal, and may still be quite expensive. Recycling is not always cheap.

The particle size is a few microns, not ideal for this application, although good performance is obtained considering that large particle size.

For HER and OER simultaneous water splitting Rh-doped SrTiO₃ decorated with Ru/CrO_x is used as HER and BiVO₄ decorated with CoO_x as OER catalyst.

The authors demonstrates the universality of the process which is good to see.

I guess some of the open questions for me are:

- effect of particle size more investigated in terms of interaction with metal matrix
- interaction semiconductor / matrix: is the enhancement of charge carrier separation more efficient in semiconductor A vs B, and if so, why?

Reviewer #3 (Remarks to the Author):

In this study, the authors have demonstrated the proof-of-concept particle-implanting -PiP technique for embedding semiconductor particles in a low-melting-point (LMP) liquid metal matrixes to produce bioinspired photoactive films for photoelectrochemical water splitting. The BiVO₄ particles were embedded into the liquid metal film, resulting in 260 times higher photoanodic current density with high stability. The authors have highlighted the advantages of the embedding-type interface over surface contact or the absence of an embedding-type interface. Furthermore, the PiP strategy has shown outstanding activity and uniformity for scalable PEC cells, effectively overcoming the "area effect." This work on photoactive film LMP metal-based PiP strategy is intriguing and can be considered for publication upon addressing the comments provided below.

1. In Fig 1c, the graphic comparing point-to-face vs three-dimensional intimate interfacial contact lacks sufficient evidence to substantiate the claim. Similarly, Fig. 1e does not conclusively demonstrate three-dimensional intimate interfacial contact; it appears more consistent with point-to-face contact. To strengthen these observations, the authors might consider supplementing additional images to rule out the possibility that this is merely a case of a single particle embedding rather than uniformity. Providing more visual evidence will enhance the clarity and robustness of the presented results.

2. The correlation between the level of interface contact and observed activity is crucial for understanding the improved charge separation efficiency resulting from uniform and higher interfacial contact between semiconductors or metal/semiconductors. How can it be conclusively determined that the interface contact between semiconductor/metal is intimate enough for a significant enhancement in activity?

3. Fig. 1d suggests that some particles may be more embedded into the metal matrix than others; is there an optimization process?

4. (Line 192) In Fig. 2f, it is stated that "the enhanced SPV signals on {010} facets exceed the signals on {011} facets." The reason for this discrepancy is not clearly explained. When BiVO₄ particles are in close surface contact with the LMP metal film (Fig. 2b), the SPV signal is significantly lower along the (010) facet. While it is understandable that embedding BiVO₄ in LMP leads to larger interface contact and higher charge separation, the reason for the increased charge separation along the (010) facet in the embedded case compared to just surface contact with LMP needs clarification. Is there a connection between this phenomenon and probable changes in the energy band of (011) facets after embedding into LMP? Will the metal-BiVO₄ interface remain the same along the (011) and (010) facets in surface contact and when embedded with LMP?

5. The comparison tables S1, S2, S3, and S4 indicate that the literature compared is not the latest, with a lack of recent articles in the comparison. Most of the cited and compared literature has reported results with Na₂SO₄, while this study used 1 M potassium borate (pH 9) as an electrolyte solution. How does the activity of the samples prepared via the PiP technique compare in Na₂SO₄ electrolyte?

6. Most of the references cited in the main manuscript are dated, with only a few references from the last 2-3 years. Are there no recent reports in this field highlighting the progress that has been made?

7. How does the PiP strategy compare with other techniques reported before? Are there any limitations associated with the PiP technique?

8. In Fig. S15, the optical image suggests that bubble formation is occurring selectively on certain locations rather than uniformly across the film. Does this imply that OWS is occurring selectively or is it facet-dependent?

Response to reviewers' comments

Reviewer #1:

This study is a tour de force on scalable photoelectrode manufacturing, which deserves to be published in a high impact journal. The study introduces a particle-implanting technique for scalable photoelectrode and photocatalytic Z-scheme fabrication, which is applied to a range of light absorbers and substrates. The study is thoroughly conducted and will provide value for many researchers in the solar fuels field. However, a few aspects remain to be addressed. The manuscript can be accepted once these points are included in the main text.

Response: Thanks for your constructive and positive comments. We have revised our manuscript point-to-point according to your suggestions.

1. How do you ensure a homogeneous surface wetting for all substrates employed? Are the pristine Si, FTO or flexible PET substrates used as purchased, or are you performing any surface treatment to improve wetting? Field metal coating can pose challenges due to surface tension, so I am wondering if the $\sim 1\ \mu\text{m}$ film thickness is homogeneous across all substrates.

Response: In response to the fact that the homogeneous coating of melted Field's metal on substrates is relative difficult due to its high surface tension, we adopted a blade to rub the liquid metal on the pristine Si, FTO or flexible PET substrates for the homogeneous surface wetting without any surface treatment. The obtained films are homogeneous across all substrates as confirmed from the SEM images (**Figure R1**).

Figure R1. The top-viewed SEM images and corresponding EDS mappings of Field's metal films coated on (a-d) Si, (e-h) FTO, and (i-l) PET substrates.

2. I don't think the PSI/PSII schematic in **Fig. 1a** is necessary. The thylakoid lipid

membrane operates quite differently from the current approach, and the inorganic oxide particles are quite different from the complex natural photosystems. As mentioned earlier, the study has plenty of merit on its own, without needing this bio-inspired comparison. The same comment applies to the **labelling of Fig. 4a**, as you are not embedding the natural PSI/PSII complexes in the Field metal, but inorganic oxide particles.

Response: The concept of “Z-scheme water-splitting system”, just mimicking the Z-scheme charge transfer path of natural photosynthesis, uses two narrow band-gap semiconductors connected in series by a charge transfer mediator to produce hydrogen and oxygen, separately. These two types of photoactive semiconductors perform similar tasks of the PSI/PSII in the photosynthesis system, while the conductive metal film serves as function as thylakoid lipid membrane for its mediator role of directional charge carrier transfer between the two photoactive units mentioned above. Although the physicochemical properties of these inorganic oxide particles and the metal film are quite different from the complex biomolecule counterparts in natural photosystems, the operation principles of these two systems for the solar-to-chemical conversion and storage are quite similar.

Furthermore, inspired by the cellular structure of chloroplast thylakoid membrane in plant cells, in which pigment-protein complexes (Photosystem I / II) are embedded in lipid matrix (Fig. 1a) for efficient photosynthesis, we proposed a facile and scalable particle-implanting (PiP) fabrication technique for embedding semiconductor particles in low-melting-point (LMP) liquid metal matrixes to produce bioinspired photoactive films. Such route not only mimics the Z-scheme charge transfer but also takes advantage of the embedded structure for efficient solar water splitting. Of course, the study has plenty of merits on its own, but it starts from an inspiration from the cellular structure of thylakoid membrane. Therefore, we just made an analogy with the thylakoid membrane but not simply replicated it. To avoid the misinterpretation, we changed the labelling in Fig. 4a, where PSI and PSII were changed into HERP (HER photocatalyst) and OERP (OER photocatalyst), respectively, according to your comment. In addition, Fig. 1a in the original version has been presented in Fig. S1 in Supplementary Materials.

3. Supplementary Fig. 6a is a key plot of the paper and should replace the current Fig 3a. The poor interface of BiVO₄ and FTO without binders is well known, hence, the focus on 260x improvement does not say much. The comparison with in situ hydrothermally grown FTO|BiVO₄ is much more appropriate and still yields a respectable performance, hence that result should be shown in Fig. 3a as mentioned above.

The authors only describe in the Methods that the particle-post-assembly technique

relates to a TiO_2 binder obtained from TiCl_4 . The wide bandgap TiO_2 binder may not have the ideal band levels to facilitate charge transfer between other oxides and FTO and may actually act as an insulating barrier. That is why the direct FTO|oxide interface of the *in situ* grown oxides provides a better comparison.

The concluding paragraph also mentions the “260 times higher photoanodic current density in photoelectrochemical water oxidation than that without embedding-type interface”. Again, this refers to a different type of interface (FTO| BiVO_4), whereas the improvement without embedding BiVO_4 powder in the Field metal is only 8x, as shown from the current Fig. 3a.

Response: Although supplementary Fig. 6a is a key plot of the paper, the quantitative comparison of photocurrent densities cannot be directly read from it. So, we extracted the key information from it as shown in Fig. 3a, hoping that this quantitative comparison is clear at a glance for readers.

We agree well with your point that the direct FTO|oxide interface of the *in situ* grown oxides might provide a better comparison, so we added the photocurrent density of the BiVO_4 photoelectrode *in situ* hydrothermally grown on FTO in Fig. 3a according to your suggestion.

In the concluding paragraph, to make the conclusion convincing and accurate, we have modified the description “260 times higher photoanodic current density in photoelectrochemical water oxidation than that without embedding-type interface” into “260 times (or 8 times) higher photoanodic current density in photoelectrochemical water oxidation than that assembled on FTO (or LMP liquid metal film) without embedding-type interface”.

4. Fig. 3d can be shown as a 2D plot, with sample numbers (1-9) on x-axis. It is hard to compare similar photocurrents on this 3D plot.

Response: We have changed Fig. 3d from the 3D plot into 2D plot for easy comparison according to your suggestion.

5. The 120 h stability of the metal| BiVO_4 photoanode is remarkable. However, Field metal can become oxidized at moderately positive potentials. So, how does the metal| BiVO_4 interface remain stable without decomposing? Does CoBi deposit on the Field metal as well at the 1.23V vs. RHE deposition potential? Supplementary Fig. 11 appears to show a flaky interface; can the authors confirm with EDX if CoBi is passivating the FM surface from oxidation?

Response: We have been cognizant of the unavoidable surface oxidation of Field's metal at moderately positive potentials, so we pre-heated this LMP metal embraced photoanode in air to form a thin oxide layer atop the metal film before testing as described in the methods. The formed oxide layer may protect the metal film from being

further oxidized, thus keeping the metal|BiVO₄ interface stable. As photogenerated holes accumulate on the BiVO₄ particle surface upon light irradiation, photoelectrochemically deposited CoBi majorly distributes on the BiVO₄ particles. However, the trace of CoBi has also been recognized on the Field's metal at the 1.23V vs. RHE deposition potential as confirmed from the SEM images and corresponding EDS mappings (Figure R2). So, the deposited CoBi might passivate the Field's metal surface together with the preformed oxide layer from further oxidation, enabling a long-term stability of the synthesized electrode.

Figure R2 (a) The top-viewed SEM images of the BiVO₄ particles-embedded photoelectrode with CoBi decoration and corresponding EDS mappings of (b) Bi, (c) In, (d) Sn, (e) O, (f) V, (g) Co and (h) B.

6. Does performance depend on surface oxides, or the work function of the low melting alloy? Fig. 4g,h indicates that it could, so more discussion would be welcome.

Response: It is absolutely true that the performance of LMP metal embraced photoactive films closely correlates with surface oxides, or the work function of the low melting alloy. In contrast to Schottky contact, Ohmic contact between the semiconductor and the LMP metal is more favorable for the efficient collection of photogenerated carriers. So, it is anticipated that the performance of photoactive films could be further improved by matching the work functions of the LMP metal and semiconductors to form an Ohmic contact. Coincidentally, an added benefit of such films is the wide range of LMP metals and semiconductors that can be used and retrieved, facilitating the construction of more efficient photoelectrodes and Z-scheme photocatalytic sheets by properly adjusting the work function of LMP metals with the Fermi levels of semiconductors. On the other hand, we pre-heated the LMP metal embraced photoanode in air to form a thin oxide layer atop the metal film, which could protect the metal film from being further oxidized during testing and thus keep the metal|BiVO₄ interface stable. The discussion on the surface properties of the components of the LMP metal embraced photoelectrode is supplemented according to your suggestion.

7. One very general, big picture question is how do the photoelectrodes and photocatalytic Z-schemes behave under heating? How can these systems prevent delamination or degradation under real-world conditions, where solar panels commonly reach over 60C under sunlight irradiation?

This delamination process is presented as recyclability in Fig. 5e, however, the photographs are too small, so the text and details are not really visible. I'd suggest the authors increase the photos, and show larger close-up images in the SI.

Response: Different from the solar panels that are directly exposed to solar irradiation in ambient condition, the photoactive films are immersed in water solution to realize solar-to-hydrogen conversion. As water has a high specific heat capacity and could be circulated during operation, the temperature of water solution is much less than 60 °C under daily solar irradiation. Therefore, the photoactive films can well bear real-world conditions without delamination or degradation issues.

According to your suggestion, we further increased the photos, and provided close-up images of the larger sized (10×10 cm) particle-embedded photoelectrode based on commercial WO₃ particles and the retrieved raw materials by ultrasonicing it in hot water bath, as shown in Fig. S29 in the revised Supplementary Materials.

8. Fig. 4h: can you mention the STH efficiencies of both Au and Field metal systems? Only the STH of the Field metal system is mentioned later on.

The Methods mention that a $\lambda \geq 420$ nm cutoff filter is used, however, both BiVO₄ and SrTiO₃ are mostly active in the UV region. Is the filter fixed within the solar light simulator? It may be more impressive to show STH values without the $\lambda \geq 420$ nm filter.

Response: The STH efficiency of Au based system has also been involved in the revised manuscript according to your suggestion. The long-term water splitting tests were carried out by using a 300 W Xe lamp equipped with a $\lambda \geq 420$ nm cutoff filter as the light source. While the STH efficiency measurement was achieved by using the AM 1.5 G sunlight simulator (100 mW cm⁻²) as the light source without the cutoff filter.

9. How does this system compare with similar large scale, flexible Ti mesh or Ti foil photoanodes (10.1002/anie.201207578, 10.1038/s41586-022-04978-6)?

Response: These large scale and flexible photoanodes (10.1002/anie.201207578, 10.1038/s41586-022-04978-6) are made of nanostructured semiconductor materials *in situ* grown on Ti mesh or Ti foil, showing relative higher photocurrent density under sunlight irradiation. However, our system shows more negative onset potentials, better active retention with scaling up, and long-term stability, which are highly demanded for practical applications. In addition, our system decouples the processes of

semiconductor synthesis and film integration, enabling efficient assembly of various semiconductors on random substrates.

10. Page 21, lines 428/429: the performance of photoelectrodes fabricated on different substrates are not shown in Fig 5a, but they should be added, as this is a key novelty point of the manuscript.

Response: We have added the performance of photoelectrodes fabricated on different substrates in Fig 5b.

Minor comments:

1. Page 14, lines 274/275: the paragraph suddenly goes from discussing performance to scalability (the scalability discussion continues in the next paragraph). It would be easier to follow if this paragraph would be split into two.

Response: We have divided the paragraph into two paragraphs according to your suggestion.

2. Page 31, lines 679/681: “A Ru (0.5 wt%)/Cr₂O₃ (0.5 wt%) core/shell cocatalyst was loaded on Rh:SrTiO₃ via a two-step photo-deposition from the solution containing RuCl₃·nH₂O and K₂CrO₄, sequentially, in vacuum.” Is this a solution, or a vacuum process?

Response: This is a solution process carried out in vacuum circumstance.

3. Page 35, lines 763/764: “a 2 μm-thick Ti layer was deposited via DC magnetron sputtering (KC-CK350), using a metal copper (purity: 99.995%, size: φ 50.8*3 mm) as the target”. Is this a Cu or Ti target?

Response: It is a Ti target rather than a Cu target. Thanks a lot for your help on correcting the error.

Reviewer #2:

The authors describe a novel method to embed semiconductor nanoparticles in liquid metal matrixes for photoelectrochemical water splitting. The performance is reported to increase drastically and this is attributed to the strong interaction between particle and metal matrix, which results in efficient charge carrier transport.

As metal matrix, the authors used the Field’s metal which is an alloy of Bi, In and Sn. In is particularly expensive. The BiVO₄ particles are prepared by hydrothermal method, which is reported by them in the introduction as not suitable for scaling up. The authors mentioned at the end using recycled materials would be a way around it, but it is not

ideal, and may still be quite expensive. Recycling is not always cheap.

Response: Thanks for your comments. As we have mentioned in our manuscript, although the LMP metal (Field's metal, Bi : In : Sn = 32.5 : 51 : 16.5 wt%) contains scarce In element, this PiP technique possesses the feature of excellent material recyclability just by applying ultra-sonication treatment in hot water, which could efficiently separate the semiconductor particles from the LMP (Field's) metal (Fig. 5e and Supplementary Fig. S29). The use of hot water and ultrasonication make the recycle process cheap and scalable. Furthermore, the photoactive films constructed from the recycled metal deliver photocatalytic overall water splitting activity and PEC performance almost identical to the fresh ones (Fig. 5f and Supplementary Fig. S30), confirming the good recyclability of the Field's metal.

We stated that hydrothermal method is not suitable for scaling up synthesis of semiconductor films rather than semiconductor powders in the introduction. Different from the rigid semiconductor films hydrothermally grown on substrates, the size of which is limited by the container, the hydrothermally synthesized particles can be piled up in the enclosed reaction container for massive synthesis. Thus, the BiVO₄ particles can be massively prepared by the hydrothermal method and then be assembled on desirable substrates via the PiP technique. In brief, the PiP-based technique proposed in this work can decouple the merit of large-scale production and good recyclability, which could provide an economy way to produce efficient photoelectrode for water splitting.

The particle size is a few microns, not ideal for this application, although good performance is obtained considering that large particle size.

Response: We adopted the micron-sized BiVO₄ particles with well-developed facets as a model material to unfold advantages of the LMP metal embraced photoactive films assembled by the PiP technique, showing impressive performance. Furthermore, we demonstrated the universality of this route, where varied semiconductor particles with different sizes can be effectively integrated on random substrates via the PiP technique. Therefore, this technique is applicable for the construction of efficient photoelectrodes and Z-scheme photocatalytic sheets by using properly sized semiconductor particles, as supported by the super performance of sub-micron-sized ZnO particle embedded photoelectrode over nanostructured ZnO films (Fig. S24 in revised Supplementary Materials).

For HER and OER simultaneous water splitting Rh-doped SrTiO₃ decorated with Ru/CrO_x is used as HER and BiVO₄ decorated with CoO_x as OER catalyst. The authors demonstrates the universality of the process which is good to see.

Response: Indeed, the PiP technique developed here has the outstanding merit of the

universality of the process in fabricating large-scale and stable photoactive films.

I guess some of the open questions for me are:

- effect of particle size more investigated in terms of interaction with metal matrix

Response: The interaction strength between semiconductor particles and metal matrix is correlated with the size and shape of particles. As the liquid metal film has high surface tension, a critical force is required to be imposed on particles for the efficient implantation into the metal film. The deeper implantation of particles, the stronger interaction with the metal matrix. The large sized particles with sharp edges are easily implanted into the metal matrix due to the stronger force implemented on them during the PiP technique. While the ultrasmall particles with smooth surface are difficult to be deeply implanted into the metal matrix, which can be ascribed to the restricted force delivered on them in the PiP technique. Note that the small sized particles are better for carrier transportation than the large sized ones. Therefore, the optimized particle size should be on the scale of hundreds of nanometers, trading off between interactions and carrier transport.

One additional merit of the large sized particles is the superior stability themselves with respect to small-sized particles.

- interaction semiconductor / matrix: is the enhancement of charge carrier separation more efficient in semiconductor A vs B, and if so, why?

Response: As different semiconductors have different electronic structures (i.e., band gaps, band edge alignments, and Fermi energy levels), Schottky (or Ohmic) contact might be formed across the metal/semiconductor interface depending on the work functions of the LMP metal and semiconductor. In contrast to Schottky contact, Ohmic contact between the semiconductor and the LMP metal is more favorable for the efficient collection of photogenerated carriers. So, the enhancement of charge carrier separation is more efficient in the semiconductor featuring Ohmic contact with the LMP metal than in the one featuring Schottky contact with the LMP metal.

Reviewer #3:

In this study, the authors have demonstrated the proof-of-concept particle-implanting - PiP technique for embedding semiconductor particles in a low-melting-point (LMP) liquid metal matrixes to produce bioinspired photoactive films for photoelectrochemical water splitting. The BiVO₄ particles were embedded into the liquid metal film, resulting in 260 times higher photoanodic current density with high stability. The authors have highlighted the advantages of the embedding-type interface over surface contact or the absence of an embedding-type interface. Furthermore, the

PiP strategy has shown outstanding activity and uniformity for scalable PEC cells, effectively overcoming the "area effect." This work on photoactive film LMP metal-based PiP strategy is intriguing and can be considered for publication upon addressing the comments provided below.

Response: Thanks for your constructive and positive comments. We have revised our manuscript point-to-point according to your suggestions.

1. In Fig 1c, the graphic comparing point-to-face vs three-dimensional intimate interfacial contact lacks sufficient evidence to substantiate the claim. Similarly, Fig. 1e does not conclusively demonstrate three-dimensional intimate interfacial contact; it appears more consistent with point-to-face contact. To strengthen these observations, the authors might consider supplementing additional images to rule out the possibility that this is merely a case of a single particle embedding rather than uniformity. Providing more visual evidence will enhance the clarity and robustness of the presented results.

Response: Actually, the top viewed SEM image (Fig. 1c in the revised manuscript) shows that BiVO₄ particles are embedded into the metal film uniformly. To further confirm the conclusion, we broken apart the film to expose the cross-section for the SEM observation. The cross-sectional SEM image demonstrates that BiVO₄ particles are uniformly embedded into the LMP metal film (Fig. R3), ruling out the possibility that it is merely a case of a single particle embedding.

Figure R3 The cross-sectional SEM image of the LMP metal embraced BiVO₄ photoelectrode.

2. The correlation between the level of interface contact and observed activity is crucial for understanding the improved charge separation efficiency resulting from uniform and higher interfacial contact between semiconductors or metal/semiconductors. How can it be conclusively determined that the interface contact between semiconductor/metal is intimate enough for a significant enhancement in activity?

Response: To conclusively demonstrate the formation of the intimate contact across semiconductor/metal interface, we cut a representative BiVO₄ particle embedded in the LMP metal matrix into a slice using focused ion beam (FIB) technique. The sectional view of FIB cutting of single BiVO₄ particle together with the LMP metal reveals an intimate interface contact between the faceted BiVO₄ and the LMP metal matrix regardless of facets (Fig. R4), which is even better than the interface formed between the BiVO₄ particle and post sputtered W metal film during the FIB technique. Therefore, the significant enhancement in activity could be attributed to the intimate interface contact.

Figure R4 The sectional view of FIB cutting of a representative particle of BiVO₄ with well-developed facets together with the LMP metal.

3. Fig. 1d suggests that some particles may be more embedded into the metal matrix than others; is there an optimization process?

Response: It is true that some particles may be more embedded into the metal matrix than others as shown in Fig. R3. We are also seeking a better process for the better embedment of particles in the LMP metal matrix. A possible way to optimize the film quality is tuning the thickness of the LMP metal film to be the half of particle size, and then applying strong enough press force to make particles penetrate the film to touch the substrate. Consequently, semiconductor particles might be uniformly half embedded into the LMP metal film.

4. (Line 192) In Fig. 2f, it is stated that "the enhanced SPV signals on {010} facets exceed the signals on {011} facets." The reason for this discrepancy is not clearly

explained. When BiVO₄ particles are in close surface contact with the LMP metal film (Fig. 2b), the SPV signal is significantly lower along the (010) facet. While it is understandable that embedding BiVO₄ in LMP leads to larger interface contact and higher charge separation, the reason for the increased charge separation along the (010) facet in the embedded case compared to just surface contact with LMP needs clarification. Is there a connection between this phenomenon and probable changes in the energy band of (011) facets after embedding into LMP? Will the metal-BiVO₄ interface remain the same along the (011) and (010) facets in surface contact and when embedded with LMP?

Response: The more holes collected at the BiVO₄ surface can be attributed to the efficient charge separation process between the BiVO₄ particles and the LMP metal film substrate. After embedding the BiVO₄ particles into the LMP metal film, the positive SPV signals on the BiVO₄ surface are further enhanced by the effective charge separation between them. As more photocarriers can be generated on {010} facets of BiVO₄ under light irradiation due to its larger exposed surface area relative to {011} facets, the efficient collection of photoelectrons between the BiVO₄ particles and the substrates may give rise to more holes left on {010} facets of BiVO₄, resulting in obvious increase of SPV signals on {010} facets, as compared to the {011} facets. That is a possible reason for the enhanced SPV signals on {010} facets exceed the signals on {011} facets.

In addition, the probable changes in the energy band structure of {011} facets after embedding into LMP might be a plausible reason as well. This suggestion inspires us to deeply investigate the system for better understanding in further study.

The metal-BiVO₄ interface remain the same along the {011} and {010} facets after being embedded into LMP metal matrix as confirmed from Fig. R4.

5. The comparison tables S1, S2, S3, and S4 indicate that the literature compared is not the latest, with a lack of recent articles in the comparison. Most of the cited and compared literature has reported results with Na₂SO₄, while this study used 1 M potassium borate (pH 9) as an electrolyte solution. How does the activity of the samples prepared via the PiP technique compare in Na₂SO₄ electrolyte?

Response: We have updated the latest literatures for comparison in Tables S1, S2, S3, and S4 according to your suggestion. The use of 1 M potassium borate (pH 9) as the electrolyte solution is because the LMP metal is stable in it during the PEC test. However, the LMP metal is unstable in Na₂SO₄ electrolyte during the PEC water oxidation test. As sulfate is a strong acid radical, the local acid environment at photoanode surface gives rise to the formation of sulfuric acid near surface during OER, resulting in the dissolution of LMP metal from electrodes. On the contrary, borate is a weak acid radical, the formed local boric acid environment cannot dissolve the LMP

metal matrix. Therefore, the PEC was tested in the potassium borate electrolyte rather than in Na₂SO₄ electrolyte.

6. Most of the references cited in the main manuscript are dated, with only a few references from the last 2-3 years. Are there no recent reports in this field highlighting the progress that has been made?

Response: We have updated the references with the recent literature reports in this field in the revised manuscript according to your suggestion.

7. How does the PiP strategy compare with other techniques reported before? Are there any limitations associated with the PiP technique?

Response: The photoactive films derived from the PiP technique show advantages of robustness, active uniformity, and scalability. In addition, the PiP technique decouples the processes of semiconductor synthesis and film integration, enabling efficient assembly of various semiconductors on various substrates. All these unique features facilitate future practical applications compared with previously reported techniques.

As the liquid metal film has high surface tension, a critical force is required to be imposed on particles for the effective implantation into the metal film. The ultrasmall particles with smooth surface are difficult to be deeply implanted into the metal matrix due to the restricted force delivered on them during the PiP process. This limitation needs to be overcome in further studies.

8. In Fig. S15, the optical image suggests that bubble formation is occurring selectively on certain locations rather than uniformly across the film. Does this imply that OWS is occurring selectively or is it facet-dependent?

Response: The bubbles are dynamically evolving from small to big and finally disappearing during the test. At a moment, the size of bubbles on different locations is variable because they are at different phases of evolution. Therefore, it looks like that bubbles are selectively produced on certain locations rather than uniformly across the film.

REVIEWERS' COMMENTS

Reviewer #1 (Remarks to the Author):

The authors have done an excellent work in responding to the reviewers' comments. However, Figures R1-R4 and the corresponding discussion from the response letter should also be integrated in the Supplementary Information and main text, as this will be useful for readers who wish to understand the PiP technique. For instance, Figure R2 and the discussion on pre-oxidizing the Field metal surface in air to prevent surface corrosion is key for the replicability of this study by other groups. This step should be described in the Methods section, and the passivating co-deposition of CoBi on the Field metal should also be mentioned in the main text.

Reviewer #2 (Remarks to the Author):

Thanks for addressing my questions about the work presented. All the concerns I had have been clarified and I am confident this manuscript will be of interest for researchers in this and related fields and happy for it to be published in Nature Commun.

Reviewer #3 (Remarks to the Author):

The authors have satisfactorily addressed the queries raised. The manuscript can be considered for publication.

Response to reviewers' comments

Reviewer #1:

The authors have done an excellent work in responding to the reviewers' comments. However, Figures R1-R4 and the corresponding discussion from the response letter should also be integrated in the Supplementary Information and main text, as this will be useful for readers who wish to understand the PiP technique. For instance, Figure R2 and the discussion on pre-oxidizing the Field metal surface in air to prevent surface corrosion is key for the replicability of this study by other groups. This step should be described in the Methods section, and the passivating co-deposition of CoBi on the Field metal should also be mentioned in the main text.

Response: Thanks for your positive advice. We have integrated Figures R1-R4 into the Supplementary Information as Figure S2, S12, S4, and S6, respectively, and put the corresponding discussions in the proper sections of the main text. Meanwhile, the pre-oxidizing the Field metal surface in air and the passivating co-deposition of CoBi on the Field metal have been described in the Methods section and in the main text.

Reviewer #2:

Thanks for addressing my questions about the work presented. All the concerns I had have been clarified and I am confident this manuscript will be of interest for researchers in this and related fields and happy for it to be published in Nature Commun.

Response: Thanks for your positive response and recognition.

Reviewer #3:

The authors have satisfactorily addressed the queries raised. The manuscript can be considered for publication.

Response: Thanks for your positive response and recognition.